# A genetic program mediates cold-warming response and promotes stress-induced phenoptosis in *C. elegans*

**Wei Jiang[1,2,3,4†], Yuehua Wei[1,4†], Yong Long[1,4,5†], Arthur Owen[6], Bingying Wang[1,4], Xuebing Wu[7], Shuo Luo[7], Yongjun Dang[2,3], Dengke K Ma[1,4*]**

[1]Cardiovascular Research Institute, University of California, San Francisco, San Francisco, United States; [2]Key Laboratory of Metabolism and Molecular Medicine, The Ministry of Education, Fudan University, Shanghai, China; [3]Department of Biochemistry and Molecular Biology, Shanghai Medical College, Fudan University, Shanghai, China; [4]Department of Physiology, University of California, San Francisco, San Francisco, United States; [5]State Key Laboratory of Freshwater Ecology and Biotechnology, Institute of Hydrobiology, Chinese Academy of Sciences, Wuhan, China; [6]Department of Molecular Cell Biology, University of California, Berkeley, Berkeley, United States; [7]Whitehead Institute for Biomedical Research, Massachusetts Institute of Technology, Cambridge, United States

**\*For correspondence:**
Dengke.Ma@ucsf.edu

†These authors contributed equally to this work

**Competing interests:** The authors declare that no competing interests exist.

**Abstract** How multicellular organisms respond to and are impacted by severe hypothermic stress is largely unknown. From *C. elegans* screens for mutants abnormally responding to cold-warming stimuli, we identify a molecular genetic pathway comprising ISY-1, a conserved uncharacterized protein, and ZIP-10, a bZIP-type transcription factor. ISY-1 gatekeeps the ZIP-10 transcriptional program by regulating the microRNA *mir-60*. Downstream of ISY-1 and *mir-60*, *zip-10* levels rapidly and specifically increase upon transient cold-warming exposure. Prolonged *zip-10* up-regulation induces several protease-encoding genes and promotes stress-induced organismic death, or phenoptosis, of *C. elegans*. *zip-10* deficiency confers enhanced resistance to prolonged cold-warming stress, more prominently in adults than larvae. We conclude that the ZIP-10 genetic program mediates cold-warming response and may have evolved to promote wild-population kin selection under resource-limiting and thermal stress conditions.
DOI: https://doi.org/10.7554/eLife.35037.001

## Introduction

Temperature shifts pervasively affect numerous biological processes in all organisms. Heat shock stimuli activate expression of many heat-shock inducible genes through the sigma-32 factor and the evolutionarily conserved transcription factor HSF (Heat Shock Factor) in bacteria and eukaryotes, respectively (*Gomez-Pastor et al., 2018*; *Yura et al., 1993*). Coordinated expression of heat shock-induced chaperone proteins facilitates cellular proteostasis and adaptation to temperature upshift (*Mahat et al., 2016*; *Solís et al., 2016*). In contrast to heat shock response, how organisms respond to cold shock is still largely unknown (*Al-Fageeh and Smales, 2006*; *Choi et al., 2012*; *Yenari and Han, 2012*; *Zhu, 2016*). Although extensive RNA expression profiling studies have identified many protein-coding genes and non-coding RNAs that are regulated by cold shock via both transcriptional and post-transcriptional mechanisms (*Al-Fageeh and Smales, 2006*; *Giuliodori et al., 2010*; *Kandror et al., 2004*; *Zhou et al., 2017*), master regulators of cold shock response and cold-regulated genes (counterparts of HSF) have long been elusive and mechanisms of cold shock response in multicellular organisms remain poorly characterized.

**eLife digest** Life on earth faces constant changes in temperature. Most warm-blooded animals like humans can maintain a fairly stable body temperature, but cold-blooded animals can experience drastic shifts in body temperature.

For example, the body temperature of the worm *Caenorhabditis elegans* can vary greatly depending on its surroundings. This species has evolved an exquisite set of temperature-sensing machineries that can react even to subtle fluctuations, which enables the worm to adjust its behaviour. However, drastic shifts in temperature can cause significant changes within the organism.

Transient exposure to heat can activate genes that help cells to repair damaged proteins, while cold shock can influence the production of proteins in the cell. Although *C. elegans* can tolerate short periods of stress, an extended exposure to extreme temperatures can kill the worm. Until now, it was not known how *C. elegans* responds to cold shock followed by warmer temperatures, also referred to as cold-warming.

To address this question, Jiang et al. created random mutations in *C. elegans* and isolated the worms that responded to cold-warming differently. The results revealed a molecular pathway that turns on genes in response to cold-warming. Jiang et al. found that two genes and their proteins, ISY-1 and ZIP-10, control which other genes are switched on or off in response to this temperature change.

When the worms were exposed to cold-warming over a long period, the pathway remained active and many of the worms died, in particular older animals. These findings suggest that this genetic program might have evolved to help younger animals survive better when stress conditions are high and food resources limited.

More work is needed to explore this new pathway and its implication in the heat-cold shock mechanisms. The affected genes are often the same across different organisms and can therefore be engineered to benefit research and medical applications in unexpected ways. For example, patients suffering a heart attack or brain injury are exposed to colder temperature to prevent the risk of tissue injuries once the blood flow goes back to normal. Therefore, the findings of this study may help us to understand how human cells respond to and are protected by low temperature.
DOI: https://doi.org/10.7554/eLife.35037.002

At the organismic level, warm-blooded mammals normally keep body temperature at about 37°C and initiate multiple homeostatic mechanisms to maintain body temperature upon exposure to hypothermia (*Bautista, 2015*; *Morrison, 2016*; *Tansey and Johnson, 2015*; *Vriens et al., 2014*). In humans, therapeutic hypothermia (32–34°C) has been widely used to treat ischemic disorders and proposed to activate multifaceted cellular programs to protect against ischemic damages (*Choi et al., 2012*; *Polderman, 2009*; *Yenari and Han, 2012*). By contrast, cold-blooded animals including most invertebrates experience varying body temperature depending on the environment, but can nonetheless elicit stereotypic behavioral, physiological and transcriptional response to chronic hypothermia or transient cold shock (*Al-Fageeh and Smales, 2006*; *Garrity et al., 2010*). Like many other types of stress, prolonged severe hypothermia can lead to the death of organisms, in most cases likely because of failure in adapting to the stress, or alternatively through stress-induced phenoptosis, namely genetically programed organismic death (*Longo et al., 2005*; *Skulachev, 1999*; *2002*). Although phenoptosis has been phenotypically documented in many cases, its evolutionary significance and genetic mechanisms remain unclear and debated (*Longo et al., 2005*; *Sapolsky, 2004*).

We previously discovered a *C. elegans* genetic pathway that maintains cell membrane fluidity by regulating lipid desaturation in response to moderate hypothermia (10–15°C) (*Fan and Evans, 2015*; *Ma et al., 2015*). Expression of the gene *fat-7*, which encodes a lipid desaturase, is transcriptionally induced by 10–15°C but not by more severe hypothermia (i.e. cold shock at 0–4°C), which impairs *C. elegans* reproduction and growth, and elicits distinct physiological and behavioral responses (*Garrity et al., 2010*; *Lyons et al., 1975*; *Ma et al., 2015*; *Murray et al., 2007*). However, as severe hypothermia arrests most of cell biological processes, strong transcriptional responses to cold shock e.g. 0–4°C likely only manifest during the organismic recovery to normal ambient temperature. We

thus hypothesize that a genetic pathway differing from that operating under moderate hypothermia exposure controls the transcriptional response to severe hypothermia/cold shock followed by warming in *C. elegans*.

In this work, we performed transcriptome profiling to first identify genes that are regulated by exposure to cold shock followed by recovery at normal temperature. We then used GFP-based transcriptional reporters in large-scale forward genetic screens to identify a genetic pathway consisting of *isy-1* and *zip-10*, the latter of which responds to cold-warming (CW) and mediates transcriptional responses to CW. Unexpectedly, we found strong *zip-10* induction promotes organismal death while deficiency of *zip-10* confers resistance to prolonged CW stress, more prominently in adults than young larvae. We propose that CW activates a ZIP-10 dependent genetic program favoring *C. elegans* phenoptosis and postulate that such programmed organismic death may have evolved to promote wild-population kin selection under thermal stress conditions.

## Results

To identify new mechanisms of *C. elegans* response to severe hypothermia, we performed RNA sequencing (RNA-seq) of wild-type *C. elegans* populations after 2 hrs exposure to 4°C cold shock followed by recovery at 20°C for 1 hr. We used such CW conditions in an attempt to identify genes that specifically and rapidly respond to CW rather than those that respond to general organismic deterioration after long cold exposure. After differential expression analyses of triplicate samples, we identified 604 genes that are significantly up- or down-regulated by such CW conditions (*Figure 1—source data 1*, *Figure 1A* and *Figure 1—figure supplement 1A*). Gene ontology analysis indicates that the CW-regulated genes are involved in biological processes including lipid metabolisms, autophagy, proteostasis and cell signaling (*Figure 1—figure supplement 1B*). We generated transgenic *C. elegans* strains in which *GFP* is driven by promoters of the top-ranked CW-inducible genes. In this work, we focus on *asp-17* as a robust CW-inducible reporter gene owing to its low baseline expression level and high-fold induction by CW, features that permitted facile isolation of full-penetrance mutants after random mutagenesis (see below) with both abnormal *asp-17p::GFP* expression and altered organismic tolerance to prolonged cold stress.

*C. elegans asp-17* encodes an aspartyl-like protease with unknown molecular functions. Like other CW-inducible genes, *asp-17* up-regulation is more prominently induced by severe than moderate hypothermia followed by recovery from cold shock (*Figure 1B and C*). Among the aspartyl-like protease family members, we found that only *asp-17* was robustly and specifically induced by CW (*Figure 1D*). The up-regulation of endogenous *asp-17* by CW can be recapitulated by an integrated GFP reporter driven by the endogenous *asp-17* promoter, indicating transcriptional regulation of *asp-17* by CW (*Figure 2—figure supplement 1A*). We varied CW treatment conditions and found that the induction of *asp-17* strictly required the warming phase after cold shock (*Figure 1E*). However, heat shock at 32°C did not increase *asp-17* expression (*Figure 1E*), consistent with previous large-scale transcriptome profiling studies in *C. elegans* (*Brunquell et al., 2016*). Single-molecule fluorescent in situ hybridization (smFISH) identified the CW-induced *asp-17* predominantly in intestinal cells (*Figure 1F*). Since CW activates numerous other genes in addition to *asp-17*, we sought to use *asp-17p::GFP* as a robust readout reporter to identify the upstream genetic pathway and transcriptional regulators that control *asp-17* induction by CW.

We performed a forward genetic screen using EMS-induced random mutagenesis of a parental strain carrying a genome-integrated *asp-17p::GFP* reporter and isolated over 30 mutants with constitutive *asp-17p::GFP* expression in the absence of CW (*Figure 2—figure supplement 1B and C*). We molecularly cloned one mutant *dma50* that exhibited fully penetrant and constitutively strong expression of *asp-17p::GFP* (*Figure 2A,B* and *Figure 2—figure supplement 1D–1F*). Compared with wild type, *dma50* strongly up-regulated *asp-17::GFP* in the intestine (*Figure 2B*). By single nucleotide polymorphism (SNP)-based linkage analysis of the intestinal *asp-17p*::GFP phenotype, we mapped *dma50* to a genetic interval on Chromosome V and used whole-genome sequencing to identify candidate causal gene mutations (*Figure 2—figure supplement 1D,E*). Based on phenocopying by feeding RNAi against the candidate genes and transformation rescue of the *asp-17p*::GFP phenotype, *dma50* defines a previously uncharacterized *C. elegans* gene *isy-1* (*Figure 2A–F* and *Figure 2—figure supplement 1F*). *isy-1* (*I*nteractor of *SY*F1 in yeast) encodes a protein with strong

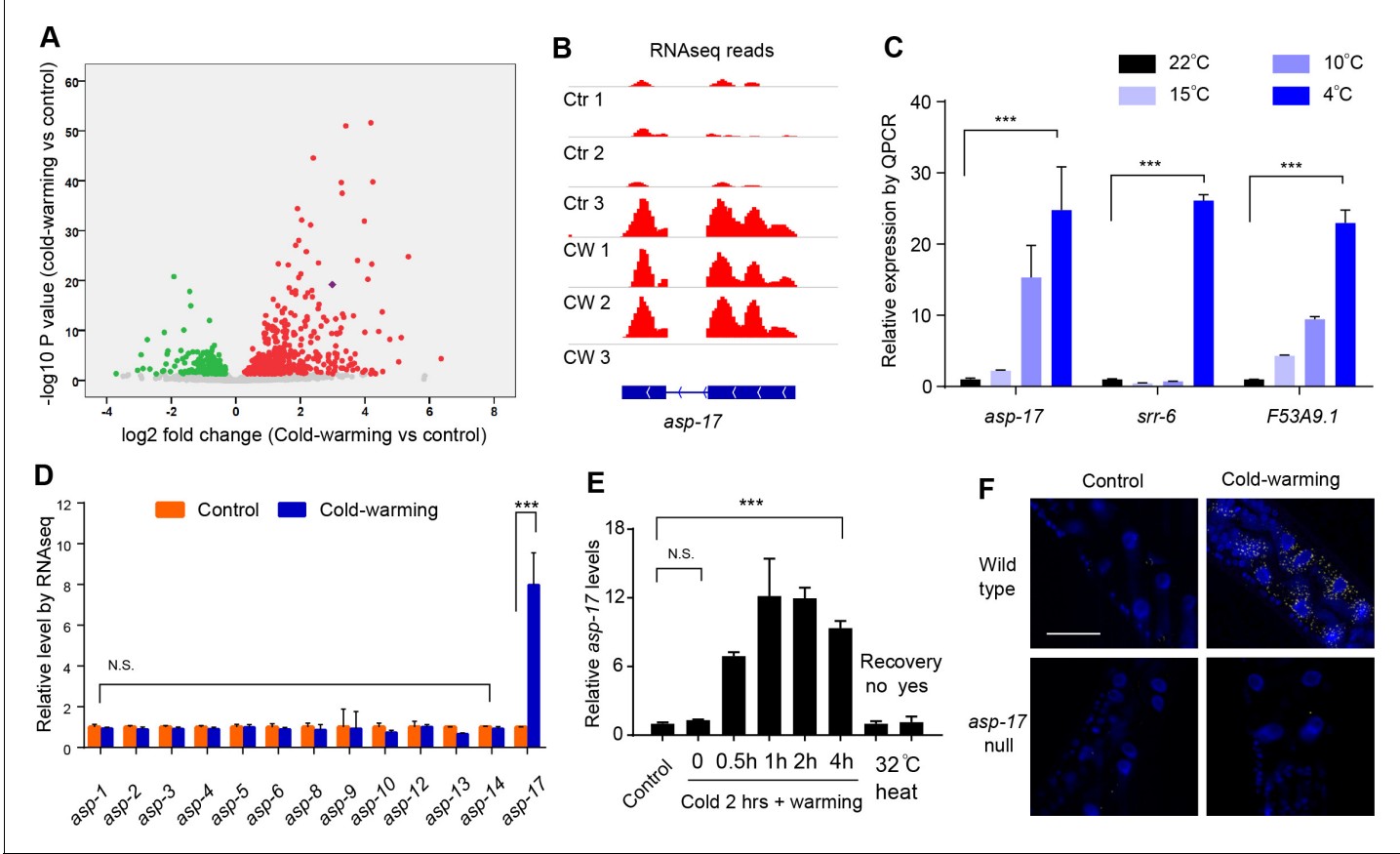

**Figure 1.** *asp-17* is robustly and specifically induced by cold-warming. (**A**) Volcano plot of RNA-seq showing differentially regulated genes (up-regulated genes in red; down-regulated genes in green; *asp-17* was indicated by purple dot) by cold-warming in wild type *C. elegans*. (**B**) RNA-seq reads at the *asp-17* locus showing consistent up-regulation of *asp-17* transcript levels in triplicate samples after cold-warming. (**C**) QPCR measurements of gene expression levels showing up-regulation of three representative CW inducible genes, including *asp-17*, after shifting from 25°C to different degrees of hypothermia (4°C, 10°C, 15°C and 22°C) lasting 2 hrs followed by recovery at 25°C for 0.5 hr. (**D**) Quantification of RNA-seq reads indicating specific up-regulation of *asp-17* but not other members of the *asp* family genes (only expressed *asp* genes are shown). (**E**) QPCR measurements of *asp-17* levels under conditions of indicated durations of cold and warming (with or without 25°C recovery for 'yes/no'). (**F**) smFISH images showing *asp-17* up-regulation (signals indicated by yellow) by CW predominantly in the intestine of wild type but not *asp-17* null animals. n ≥ 20 total animals for each group with N ≥ 3 independent biological replicates; *** indicates p<0.001. Scale bar: 50 μm.

DOI: https://doi.org/10.7554/eLife.35037.003

The following source data and figure supplement are available for figure 1:

**Source data 1.** Lists of genes up- and down-regulated by CW with adjusted p<0.05 and log2FoldChange from biological triplicate samples of wild-type *C. elegans*.

DOI: https://doi.org/10.7554/eLife.35037.005

**Figure supplement 1.** RNA-seq identified genes up-regulated by cold-warming.

DOI: https://doi.org/10.7554/eLife.35037.004

sequence similarity to an evolutionarily highly conserved family of RNA-binding proteins in eukaryotes (*Figure 2A* and *Figure 2—figure supplement 2A–C*) (*Dix et al., 1999*; *Du et al., 2015*).

*dma50* caused substitution of a negatively charged glutamate, which is completely conserved in the ISY protein family, to a positively charged lysine in the predicted coiled-coil region of *C. elegans* ISY-1 (*Figure 2A*). An *isy-1p::isy-1::GFP* translational reporter indicated a rather ubiquitous distribution of ISY-1::GFP in many tissues including intestinal nuclei (*Figure 2C*). The strong intestinal *asp-17p*::GFP expression caused by *dma50* was fully rescued by transgenic expression of wild-type *isy-1* (+), single-copy integration of a *mCherry*-tagged *isy-1(+)* allele, or *isy-1(+)* expression driven by the intestine-specific *ges-1* promoter (*Figure 2E and F*). In addition, the *ges-1*-driven transgenic expression of sense plus antisense *isy-1* RNAi fully recapitulated the *dma50* phenotype (*Figure 2D*).

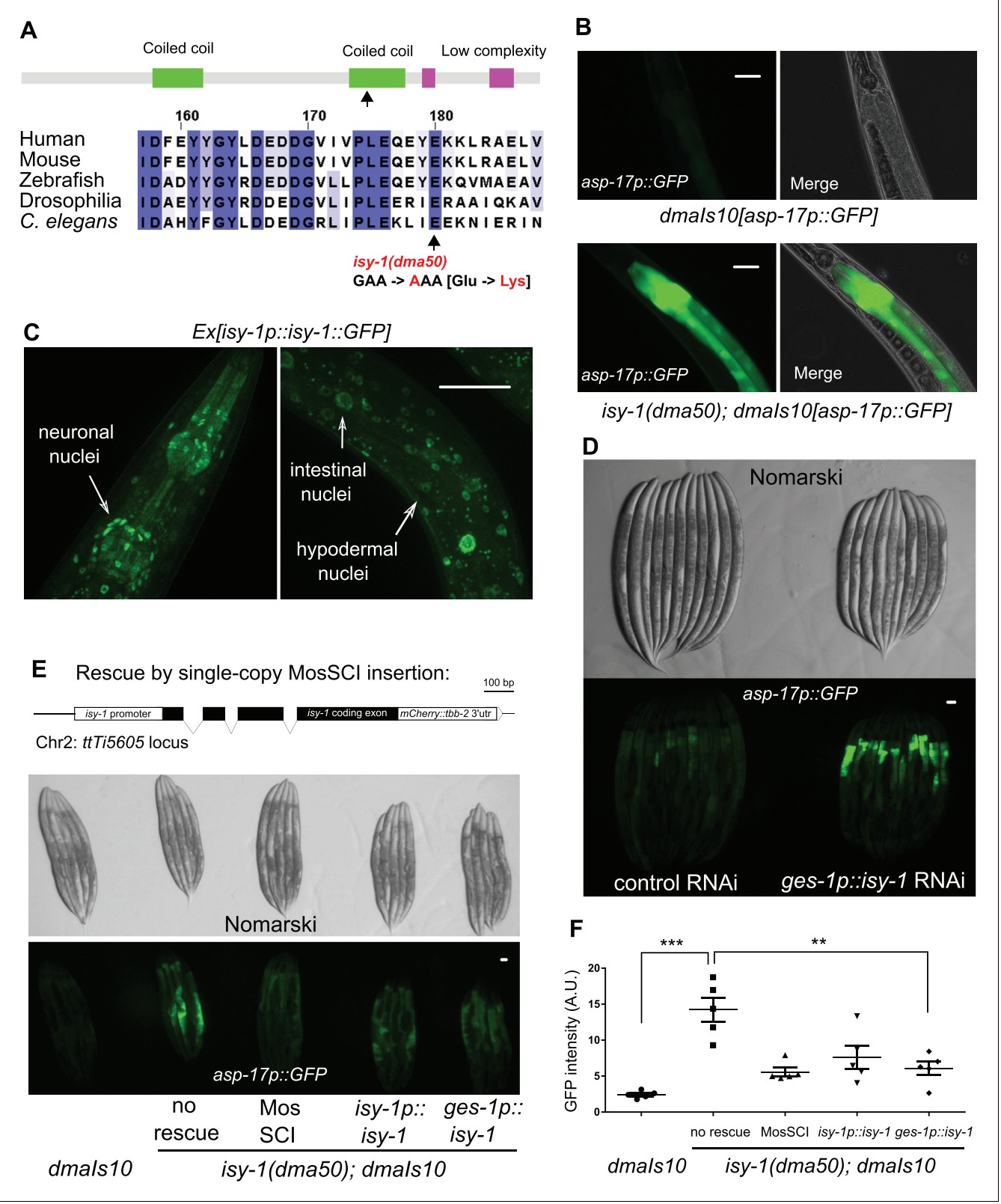

**Figure 2.** A forward genetic screen identifies *C. elegans isy-1* as a causal regulator of *asp-17*::GFP expression. (**A**) Schematic of the *C. elegans* ISY-1 protein showing the domain structure predicted by SMART (top) (http://smart.embl-heidelberg.de) and a multiple sequence alignment of ISY-1 homologues from major metazoans showing the conservation of the glutamate residue substituted to lysine by the *dma50* mutation isolated from EMS screens. (**B**) Nomarski and fluorescence images showing the phenotype of intestinal *asp-17p*::GFP in wild type and *dma50* mutants. (**C**) Fluorescence

*Figure 2 continued on next page*

*Figure 2 continued*

images showing the distribution of ISY-1::GFP driven by the endogenous *isy-1* promoter. Arrows indicate neuronal, hypodermal and intestinal nuclei. (D) Nomarski and fluorescence images showing intestinal *asp-17p*::GFP in a transgenic strain expressing RNAi against *isy-1* specifically in intestine. (E) Schematic of the *C. elegans isy-1* gene with mCherry tagged at the C-terminus (top); Nomarski and fluorescence images showing rescue of *dma50* by various transgenes (below). (F) Quantification of fluorescence intensities showing rescue of *dma50* in *asp-17p*::GFP activation. N ≥ 3 independent biological replicates; *** indicates p<0.001. Scale bar: 20 μm.

DOI: https://doi.org/10.7554/eLife.35037.006

The following figure supplements are available for figure 2:

**Figure supplement 1.** EMS screens identified *isy-1* as a regulator of *asp-17*.

DOI: https://doi.org/10.7554/eLife.35037.007

**Figure supplement 2.** ISY-1 is a *C. elegans* member of evolutionarily conserved protein family and its level is not regulated by cold-warming.

DOI: https://doi.org/10.7554/eLife.35037.008

Endogenous expression of *asp-17* was also drastically up-regulated in *isy-1* mutants (*Figure 2—figure supplement 2D*). Thus, these results identify *isy-1* as a causal cell-autonomous regulator of *asp-17*.

Human ISY1 is critical for certain microRNA processing while yeast ISY1 is a likely component of the spliceosome (*Dix et al., 1999*; *Du et al., 2015*; *Galej et al., 2016*). We found that CW-induced *asp-17* up-regulation was further enhanced in *isy-1* mutants compared with wild type (*Figure 3A*), suggesting that ISY-1 normally restricts transcriptional activity of *asp-17*. To determine the mechanism by which ISY-1 regulates transcription of *asp-17*, we sought to identify transcription factors (TF) that meet two criteria: a), its mRNA or protein products are altered in *isy-1* mutants, and b), it is genetically epistatic to *isy-1*, that is, its loss-of-function (LOF) can suppress *isy-1* LOF (thus also likely required for *asp-17* induction by CW). We performed RNA-seq from triplicate samples of wild-type hermaphrodites and *isy-1* mutants, from which we analyzed differentially expressed TF-encoding genes in *isy-1* mutants and found that a bZIP-type transcription factor-encoding gene *zip-10* met both criteria (*Figure 3B*, *Figure 3—source data 1*). *zip-10* mRNA was drastically up-regulated in *isy-1* mutants, whereas levels of closely related bZIP family genes, such as *zip-11*, were unaffected (*Figure 3C*). Importantly, genetic deletion of *zip-10* completely abrogated the ability of *isy-1* RNAi to activate *asp-17p*::GFP (*Figure 3D*). These results indicate that ISY-1 regulates *asp-17* by controlling the level of *zip-10* mRNAs.

Next, we examined how the ISY-1/ZIP-10/ASP-17 pathway is regulated by CW. CW did not apparently alter levels of endogenous *isy-1* mRNAs or mCherry-tagged ISY-1 proteins under the endogenous *isy-1* promoter (*Figure 2—figure supplement 2D and E*). By contrast, we found that CW induced drastic up-regulation of ZIP-10 proteins from a tagged *zip-10p::zip-10::EGFP::FLAG* allele in an integrated transgenic strain (*Figure 3E*). Although EGFP fluorescence was invisible in animals carrying such transgenes (likely because it is sandwiched by *zip-10* and FLAG), the striking induction of ZIP-10::EGFP::FLAG was completely blocked by RNAi against *zip-10* or *GFP*, confirming the transgene specificity (*Figure 3F*). The baseline level of ZIP-10::EGFP::FLAG was close to the detection limit of western blot under normal conditions, but nonetheless is strongly up-regulated upon RNAi against *isy-1* (*Figure 3F*). Similar to that of *asp-17*, the induction of *zip-10p::zip-10:: EGFP::FLAG* strictly required the warming phase of CW and occurred rapidly but transiently after warming during CW (*Figure 3G*). CW strongly up-regulated *asp-17* expression in both wild type and *isy-1* mutants, which exhibited abnormally high *zip-10* mRNA levels (*Figure 3H*). Furthermore, *zip-10* deletion completely abrogated the up-regulation of *asp-17* levels by CW (*Figure 3I*). We also examined the ZIP-10 dependency of other CW-inducible genes identified by RNA-seq and found that at least *cpr-3* also required ZIP-10, but other CW-inducible genes including *srr-6* and *F53A9.1*, did not (*Figure 3I*). These results demonstrate that ISY-1 suppresses *asp-17* by decreasing *zip-10* levels whereas CW up-regulates ZIP-10 protein abundance to promote *asp-17* expression.

How is *zip-10* regulated by ISY-1 and CW? Loss of ISY-1 function affected neither general intron splicing, based on an intronic GFP reporter assay, nor specific splicing of *zip-10*, although both CW and *isy-1* mutations strongly up-regulated *zip-10* mRNA levels (*Figure 3—figure supplement 1A–1E*). We constructed a GFP transcriptional reporter driven by the endogenous *zip-10* promoter and found it was markedly up-regulated by *isy-1* RNAi (*Figure 3—figure supplement 2A*). While non-thermal stresses such as hypoxia and starvation did not increase ZIP-10 levels, CW drastically

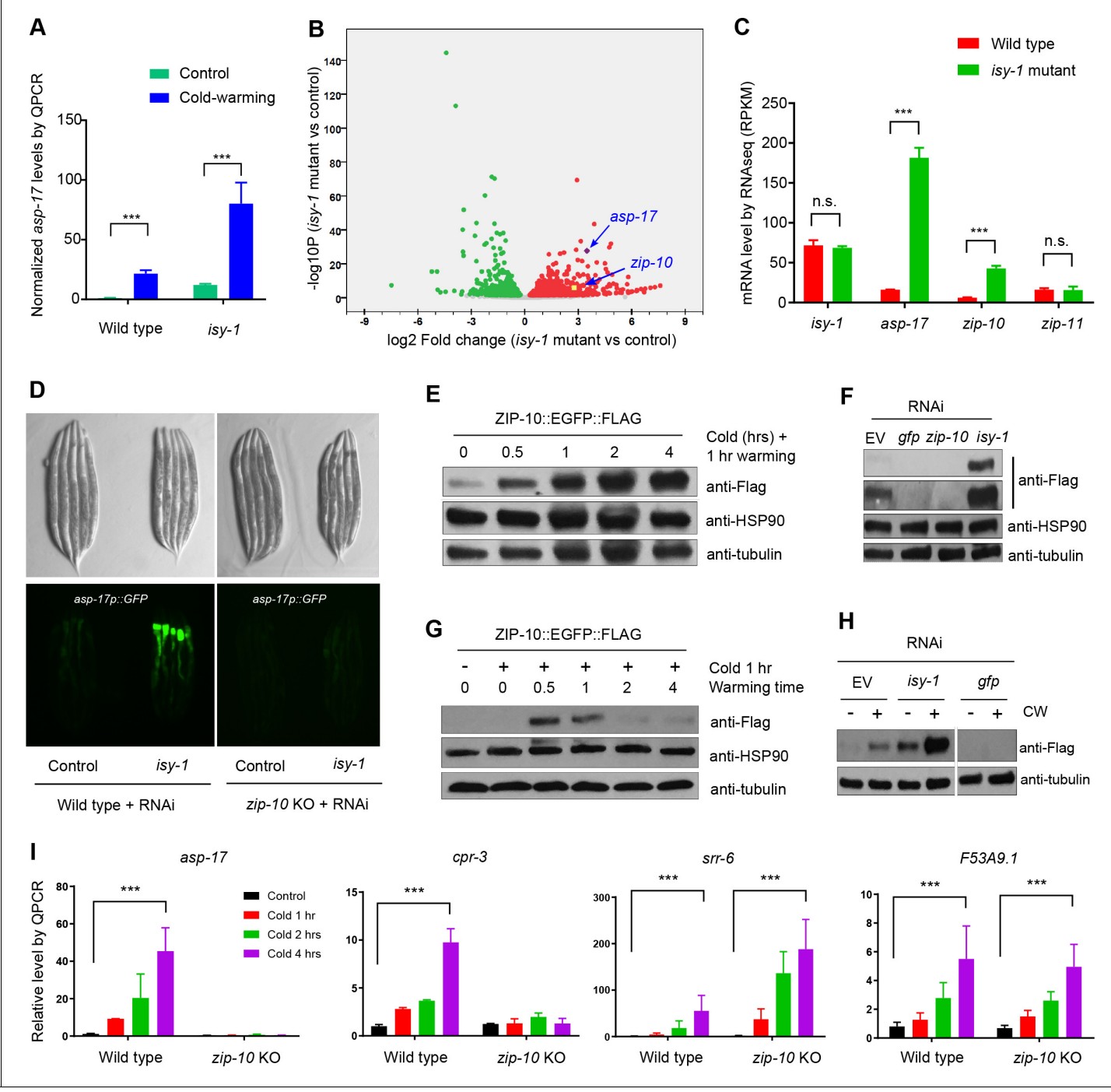

**Figure 3.** ZIP-10 acts downstream of ISY-1 and mediates transcriptional response to CW. (**A**) QPCR measurements of *asp-17* levels induced by CW in wild type and *isy-1(dma50)* mutants. (**B**) Volcano plot of RNA-seq showing differentially regulated genes (up-regulated genes in red; down-regulated genes in green) in *isy-1* mutants compared with wild type. (**C**) RNA-seq measurements of expression levels for indicated genes in wild type and *isy-1 (dma50)* mutants. (**D**) Nomarski and fluorescence images showing *asp-17p::GFP* induction by *isy-1* RNAi was blocked in *zip-10* mutants. (**E**) Western blots of the integrated *zip-10p::zip-10::EGFP::FLAG* strain showing time-dependent protein induction by CW. (**F**) Western blots of the integrated *zip-10p::zip-10::EGFP::FLAG* strain showing its up-regulation by *isy-1* RNAi and down-regulation by *GFP* or *zip-10* RNAi. Both short- and long-exposure blots are shown. (**G**) Western blots of the integrated *zip-10p::zip-10::EGFP::FLAG* strain showing its up-regulation strictly required warming after cold shock. (**H**) Western blots of the integrated *zip-10*p::*zip-10::EGFP::FLAG* strain showing its up-regulation by CW was further enhanced by *isy-1* RNAi. (**I**) QPCR measurements of gene expression levels showing ZIP-10 dependent up-regulation of *asp-17* and *cpr-3* but not *srr-6* or *F53B9.1* after cold for indicated durations and 1 hr warming. n ≥ 20 total animals for each group with N ≥ 3 independent biological replicates; *** indicates p<0.001. Scale bar: 20 μm.

*Figure 3 continued on next page*

*Figure 3 continued*

DOI: https://doi.org/10.7554/eLife.35037.009

The following source data and figure supplements are available for figure 3:

**Source data 1.** Lists of genes up- and down-regulated by the *isy-1(dma50)* mutation with adjusted p<0.05 and log2FoldChange from biological triplicate samples of wild-type and *isy-1(dma50)* mutant *C. elegans*.

DOI: https://doi.org/10.7554/eLife.35037.012

**Figure supplement 1.** ISY-1 does not affect general splicing of a GFP reporter nor specific splicing of *zip-10*.

DOI: https://doi.org/10.7554/eLife.35037.010

**Figure supplement 2.** Mechanisms of regulation and function of *zip-10*.

DOI: https://doi.org/10.7554/eLife.35037.011

increased abundance of ZIP-10 in both cytosol and nucleus without affecting abundance of other house-keeping proteins, including HSP90, tubulin and histone H3 (*Figure 3—figure supplement 2B and C*). CW up-regulation of ZIP-10 required warming and was enhanced by more prolonged cold shock (*Figure 3—figure supplement 2D*). Since CW can markedly increase *zip-10* mRNA levels but to a lesser extent than the *isy-1* mutation (*Figure 3—figure supplement 1D and E*), we tested whether ZIP-10 proteins might be regulated by CW through translational control and mRNA stability. RNAi against genes encoding eIF5 and a component of the Ccr4-Not complex did not apparently alter ZIP-10 levels (*Figure 3—figure supplement 2E*). Together, these results indicate that CW and ISY-1 regulate *zip-10* primarily at the transcriptional level.

Human ISY1 facilitates the processing of primary transcripts encoding certain families of microRNAs (*Du et al., 2015*). Both *zip-10* and *asp-17* are up-regulated in a mutant *C. elegans* strain deficient in the microRNA *mir-60* (*Kato et al., 2016*). We thus tested whether *mir-60* mediates the regulation of *zip-10* by ISY-1. Immunoprecipitation of mCherry-tagged ISY-1 followed by quantitative PCR (QPCR) revealed specific binding of primary transcripts encoding *mir-60* as well as a protein-coding gene *cebp-1* (*Figure 4A–4C*). Although neither *isy-1* nor *mir-60* levels were affected by CW, we found CW slightly increased *mir-60* binding to ISY-1, perhaps as a feedback mechanism to limit over-activation of *zip-10*-dependent genes after CW treatment (*Figure 4B*). Importantly, mature *mir-60* levels were drastically decreased in *isy-1* mutants while loss of *mir-60* led to up-regulation of *zip-10* and *zip-10*-dependent subset of CW-inducible genes, including *asp-17* and *cpr-3*, but not many other CW-inducible genes (*Figure 4D and E* and *Supplementary file 1*). The 3' untranslated region (Utr) of *zip-10* appeared not to be regulated by CW or *isy-1* RNAi (*Figure 4F*). However, *isy-1* RNAi caused an abnormally high baseline level of ZIP-10 in the absence of CW and enabled further heightened ZIP-10 up-regulation in response to CW, followed by its down-regulation over an extended period of warming (*Figure 4G*). These results indicate that CW regulates transcription of *zip-10* (and thereby that of *asp-17*), while ISY-1 controls expression of *zip-10* via *mir-60*, likely through microRNA processing and regulation of additional upstream transcriptional *zip-10* regulators that respond to CW.

We compared the genes differentially regulated by CW and those by *isy-1(dma50)* mutants and found 246 genes, including the two ZIP-10-dependent targets *asp-17* and *cpr-3*, that are commonly regulated by both conditions (*Figure 5A*, *Figure 5—source data 1*). Global transcriptome changes between these two conditions are also significantly correlated (*Figure 5B*) (correlation coefficient $R$ as 0.54, significance $P$ value as 0). We used the bioinformatics tool MEME (*Bailey et al., 2009*) to identify motifs present in the promoters (~600 bp upstream of transcription start sites) of the commonly regulated gene subset and identified a single enriched motif characterized by AT-rich sequences (*Figure 5C*). The gene most enriched with this motif is *asp-17*, the promoter of which contains 16 such motifs (*Figure 5C*). ZIP-10 is a bZIP-type transcription factor predicted to contain N-terminal low sequence-complexity domains and a C-terminal DNA-binding and glutamine-rich transactivation domain (*Figure 3—figure supplement 2F–H*). To test whether the *asp-17* promoter with the identified AT-rich motifs can be bound directly by ZIP-10, we performed chromatin immunoprecipitation (ChIP) experiments and detected *asp-17* promoter sequences in the FLAG-tagged ZIP-10 chromatin complex only under CW conditions (*Figure 5D*). These results indicate that ZIP-10 directly binds to and activates the *asp-17* promoter in the genetic program regulated by ISY-1 and CW.

The striking regulation of *zip-10* and *asp-17* by CW and ISY-1 prompted us to examine the organismic phenotype of various mutants upon prolonged CW stress. A majority of wild-type *C. elegans*

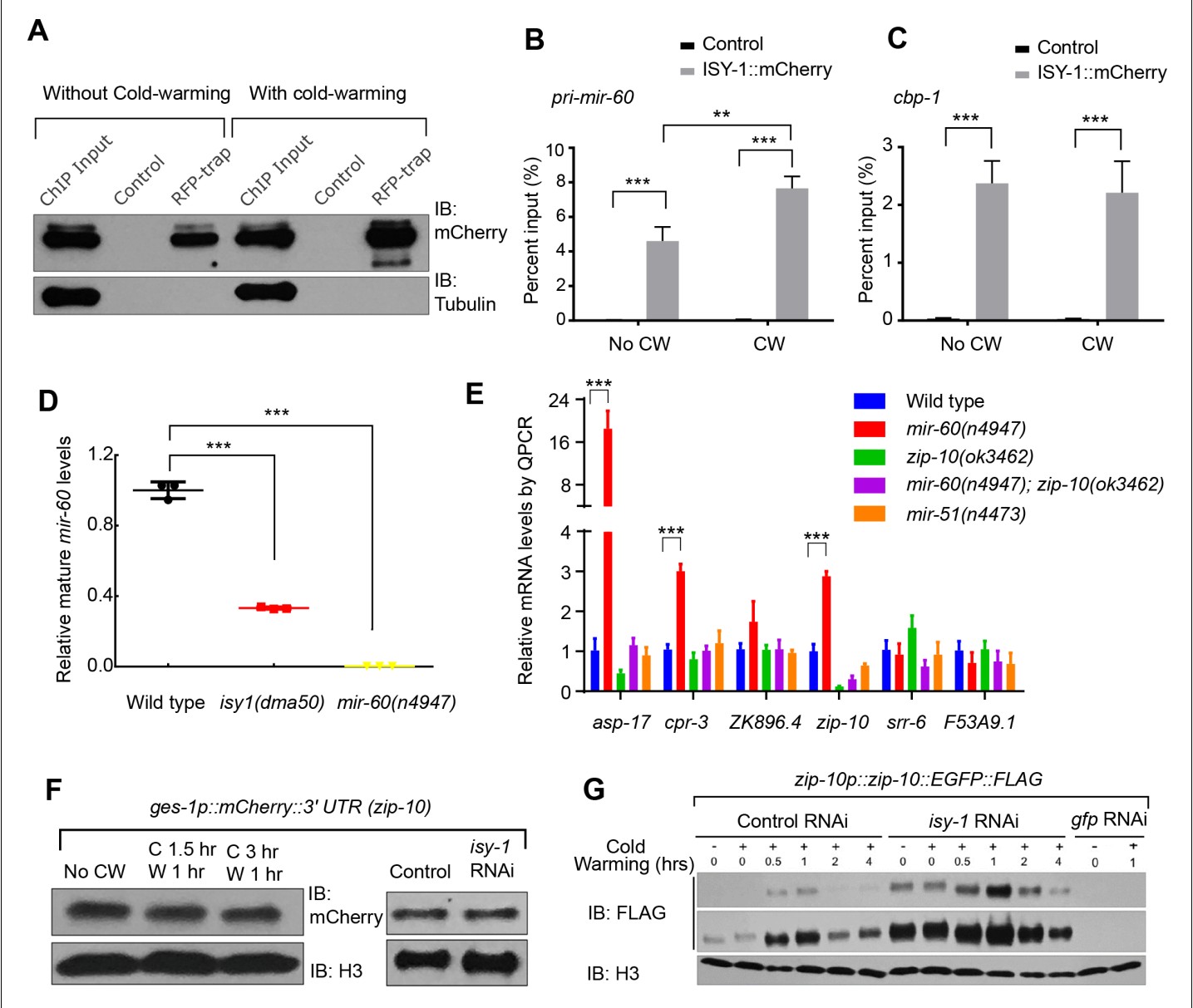

**Figure 4.** ISY-1 regulates *zip-10* via *mir-60*. (**A**) Western blot of mCherry/RFP-trapped RNA immunoprecipitates in animals treated with or without cold-warming. (**B**) QPCR measurements of the percent input for primary *mir-60* transcripts from mCherry/RFP-trapped RNA immunoprecipitates in animals treated with or without cold-warming. (**C**) QPCR measurements of the percent input for *cbp-1* transcripts from mCherry/RFP-trapped RNA immunoprecipitates in animals treated with or without cold-warming. (**D**) QPCR measurements of the mature *mir-60* transcript levels from wild type, *isy-1(dma50)* and *mir-60(n4947)* deletion mutants. (**E**) QPCR measurements of the levels of CW-inducible gene transcripts in animals with indicated genotypes and conditions. (**F**) Western blot of lysates from animals carrying the array *ges-1p::mCherry::3'utr(zip-10)* reporters with CW or *isy-1* RNAi. No change of reporter activity was observed. (**G**) Western blot of lysates from animals carrying *zip-10p::zip-10::EGFP::FLAG* reporters with various indicated CW and RNAi conditions. n ≥ 20 total animals for each group with N ≥ 3 independent biological replicates; *** indicates p<0.001; ** indicates p<0.01.
DOI: https://doi.org/10.7554/eLife.35037.013

adults died upon prolonged CW stress (e.g. 2–4°C for over 24 hrs) (*Ohta et al., 2014*). We found that *asp-17* or *zip-10* loss-of-function mutants exhibited markedly higher survival rates than wild type under the same prolonged CW stress condition (*Figure 5E*). Consistent with a role of wild-type *zip-10* in promoting organismic death, inducible *zip-10* over-expression by mild transient heat shock, mediated by the *hsp-16* promoter, promoted animal death even in the absence of CW (*Figure 5F*). By contrast, other ectopically induced *zip* genes including *zip-11* and *zip-2* did not affect animal

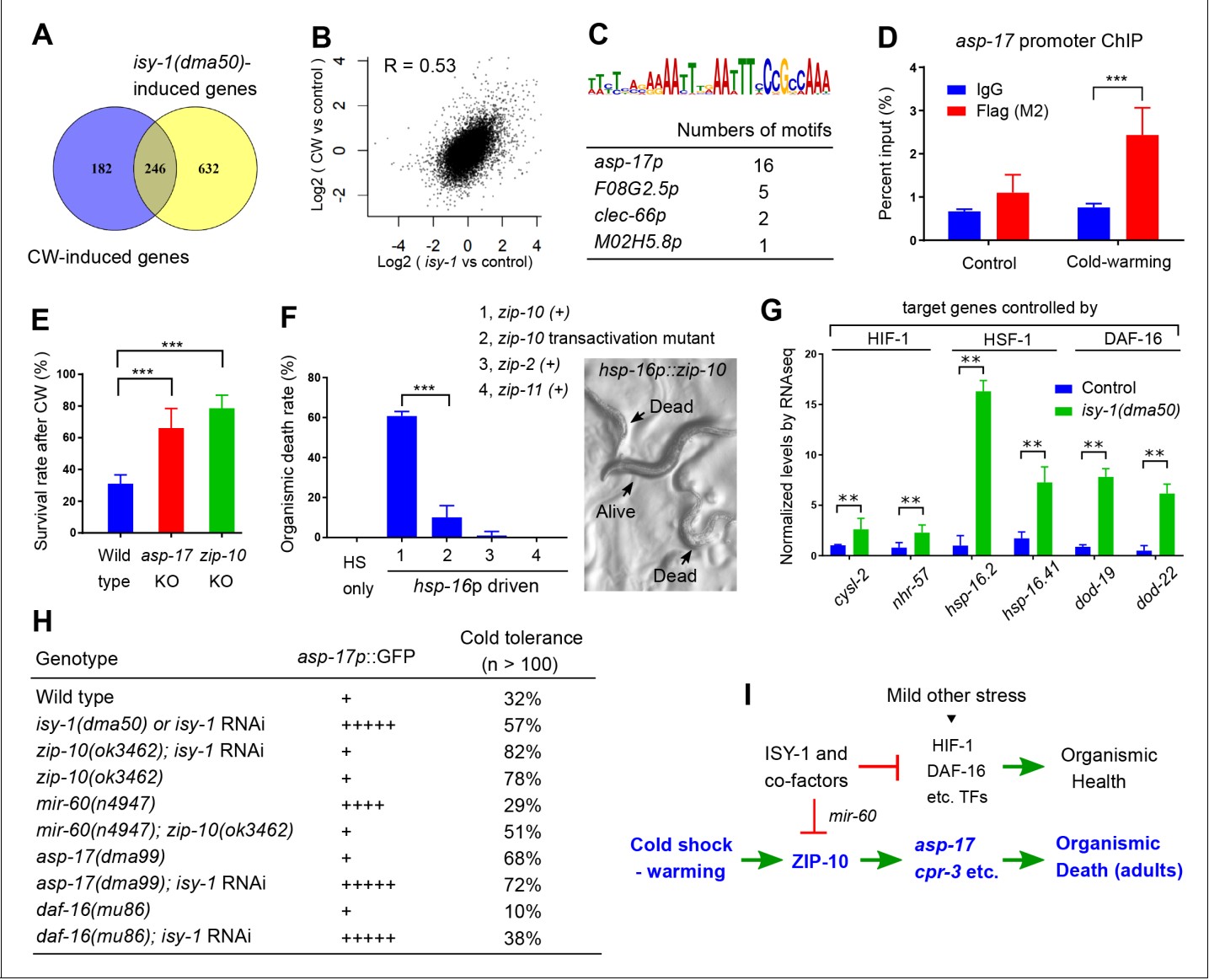

**Figure 5.** ISY-1 and ZIP-10 regulate a genetic program to promote organismic death. (**A**) Venn diagram indicating numbers of genes commonly regulated by CW and *isy-1(dma50)* mutants. (**B**) Scatter plot depicting the correlation between the transcriptome response to cold (y-axis) and the transcriptome response to *isy-1* mutation (x-axis). Shown are the log2 fold changes compared with corresponding controls. Pearson correlation coefficient and the associated P-value were calculated using R functions. (**C**) AT-rich motif identified by MEME enriched among the CW and *isy-1 (dma50)* regulated genes, with a table listing numbers of the motif present in top-ranked four genes. (**D**) ChIP-QPCR measurements of ZIP-10::FLAG binding to the *asp-17* promoter. (**E**) Survival rates of indicated genotypes after prolonged CW (4°C for 48 hrs followed by 4 hrs of warming). (**F**) Organismic death rates of indicated genotypes after heat shock (32°C) induction of *zip-10* wild type, mutant with defective transactivation C-terminus, *zip-2* and *zip-11* (left), without CW. Nomarski image (right) indicates morphologies of normal and dead animals with induction of *zip-10*. (**G**) RNA-seq measurements of gene targets of indicated TFs. (**H**) Table showing the *asp-17p::GFP* and cold tolerance phenotypes of animals with indicated genotypes. (**I**) Model for the role and regulation of the ZIP-10 pathway. n ≥ 20 total animals for each group with N ≥ 3 independent biological replicates; *** indicates p<0.001.

DOI: https://doi.org/10.7554/eLife.35037.014

The following source data and figure supplements are available for figure 5:

**Source data 1.** Lists of genes commonly regulated by CW and the *isy-1(dma50)* mutation with adjusted p<0.05 and log2FoldChange from biological triplicate samples of wild-type and *isy-1(dma50)* mutant *C. elegans*.
DOI: https://doi.org/10.7554/eLife.35037.017
**Figure supplement 1.** Identification of differentially regulated microRNAs in wild type and *isy-1(dma50)* mutants.
DOI: https://doi.org/10.7554/eLife.35037.015

*Figure 5 continued on next page*

*Figure 5 continued*

**Figure supplement 2.** ZIP-10 constitutes a genetic program promoting kin selection of *C. elegans* under resource-limiting and stress conditions.

DOI: https://doi.org/10.7554/eLife.35037.016

death, while a mutation specifically disrupting the glutamine-rich transactivation domain of ZIP-10 abolished the death-promoting effect (*Figure 5F* and *Figure 3—figure supplement 2H*). Although *zip-10* is genetically epistatic to *isy-1* in the regulation of *asp-17*, we found that *isy-1* mutants are also markedly resistant to prolonged cold stress. This paradox was resolved after we observed that many downstream target genes of the stress-coping transcription factors HIF-1, HSF-1 and DAF-16 are up-regulated in *isy-1* mutants, and LOF of at least *daf-16* could partly suppress cold tolerance by *isy-1* RNAi (*Figure 5G and H*). Since ISY-1 regulates *zip-10* via *mir-60* (*Figure 4*) supporting a role of ISY-1 in specific microRNA processing (*Du et al., 2015*), we performed small RNA library sequencing of wild type animals and *isy-1* mutants and identified specific members of microRNAs that were differentially regulated, including *mir-60* and additional microRNAs predicted to target stress-coping TFs (*Figure 5—figure supplement 1A–F*). Thus, *isy-1* mutants likely exhibit pleiotropic phenotypes caused by abnormal activation of multiple TFs in addition to ZIP-10. In contrast to ZIP-10 dependent genes (*asp-17* and *cpr-3*), the HIF-1/HSF-1/DAF-16 target genes were not apparently induced by CW (*Figure 1—source data 1*). Furthermore, unlike HSF-1 or DAF-16 that are induced by other types of stress stimuli, ZIP-10 is more strongly induced by CW in adults than in larvae (*Figure 5—figure supplement 2C*), suggesting phenoptosis-promoting effects of *zip-10* more specifically for adults. Indeed, the phenotypic difference in cold tolerance between wild type animals and *zip-10* mutants manifested more prominently in developmentally more mature-stage and older animals (*Figure 5—figure supplement 2D*). These results indicate that CW specifically activates a ZIP-10-driven and developmental stage-modulated transcriptional genetic program to promote the organismic death, or phenoptosis, of *C. elegans* (*Figure 5I*).

## Discussion

From a genetic screen for *C. elegans* mutants with altered transcriptional response to CW, we identified *isy-1* and subsequently discovered the CW and ISY-1-regulated transcription factor ZIP-10 as a key mediator of the transcriptional response to CW. A thermal stress-responding TF might be expected to promote adaptation of animals towards the stressor, causing its LOF mutants to be sensitive to the stress. Unexpectedly, we found *zip-10* mutants are markedly resistant to prolonged cold stress. However, unlike other stress-responding TFs that activate genes largely beneficial for physiological homeostasis and thus animal health under stress conditions (*Baird et al., 2014*; *Dempersmier et al., 2015*; *Hwang and Lee, 2011*; *Kandror et al., 2004*; *Kumsta et al., 2017*; *Landis and Murphy, 2010*), identified transcriptional targets of ZIP-10 include at least two Cathepsin-type proteases, CPR-3 and ASP-17 (*Figure 4E*). In contrast to aspartyl-type proteases which are largely unknown in cellular functions, caspase-type proteases are well-known apoptotic cell death executioners while CPR-4, a Cathepsin CPR-3 paralogue, has been shown to inhibit cell deaths in *C. elegans* (*Metzstein et al., 1998*; *Peng et al., 2017*; *Peter, 2011*). Ectopic expression of *zip-10* and its targets promotes organismic deaths, in contrast to the effect of *zip-10* or *asp-17* deficiency on cold tolerance (*Figure 5E and F*). As duration of cold shock affects levels of ZIP-10 and transient CW does not trigger phenoptosis, the pro-death role of the *zip-10* genetic program likely depends on multiple factors, including the duration and severity of cold exposure. Notably, apoptotic cell death-promoting effects have also been described for specific members of mammalian bZIP TFs (*Chüeh et al., 2017*; *Hartman et al., 2004*; *Ritchie et al., 2009*). The specific and robust induction of ZIP-10 by CW, the opposing cold-tolerance phenotypes caused by *zip-10* loss-of-function and gain-of-function genetic manipulations, as well as the pro-death roles of ZIP-10 targets support the notion that the *zip-10* pathway is activated by severe CW to promote phenoptosis.

How do ISY-1 and CW regulate the *zip-10* pathway? We found that the *zip-10* promoter activity responds to the loss of ISY-1, which normally maintains *mir-60* levels and thereby regulates *zip-10* transcription likely through the processing of small RNAs. Severe cold stress also leads to accumulation of another class of small RNA risiRNA, which is important for maintaining rRNA homeostasis (*Zhou et al., 2017*). Whether ISY-1 might also affect risiRNA processing remains to be characterized.

Constitutive up-regulation of ZIP-10 targets in *isy-1* mutants and the lack of evidence for regulation of ISY-1 by CW supports ISY-1 as a gate-keeper for the ZIP-10-driven transcriptional response to CW (*Figure 5I*). Regulation of *zip-10* is primarily transcriptional based on evidence we present in this study; further studies are required to discern to what extent *mir-60* might directly act at the *zip-10* locus or more indirectly impact the transcription of *zip-10*, e.g. by post-transcriptionally inhibiting translation of a transcriptional activator. Up-regulation of the activity of the *zip-10* promoter by CW indicates that additional cold-responding sensors and effectors upstream of ZIP-10 remain to be identified, by signaling mechanisms perhaps similar to the well-characterized cold-responding pathways found in other organisms (*Dempersmier et al., 2015*; *Kandror et al., 2004*; *Zhu, 2016*). Precisely how *zip-10* is regulated by CW in coordination with ISY-1 to promote *C. elegans* death under prolonged CW stress awaits further investigation.

The roles of ZIP-10 and a dedicated genetic program in promoting organismic death are surprising but would make sense in light of the evolutionary kin selection theory. Kin selection refers to the evolutionary process promoting the reproductive success of an organism's kin despite a cost to the organism's own reproduction (*Hamilton, 1963*; *Smith, 1964*). Dedicated genetic programs may have evolved to promote kin selection at the population level. Although the concept and potential mechanisms of programed organismic death, or phenoptosis, are debated, examples of kin selection and stress-induced organismic deterioration have been widely documented in many organisms (*Longo et al., 2005*; *Sapolsky, 2004*; *Skulachev, 1999*; *2002*).

Laboratory conditions for hermaphroditic *C. elegans* clearly no longer exert selection pressure for genetic programs underlying phenoptosis or kin selection. However, our mathematic modeling of an exemplar situation of population growth for wild-type and *zip-10* deficient animals under food-limiting and CW stress conditions supports the phenoptosis or kin selection hypothesis for the *zip-10* pathway (*Figure 5—figure supplement 2A and B*). Experimentally, we found that both the CW-induced *zip-10* expression and the death-promoting effect of ZIP-10 occurred more prominently in older adults than in larvae (*Figure 5—figure supplement 2C and D*). Extending from the kin selection theory, we postulate that the evolutionary advantage of programmed organismic death might manifest in the wild, where resources for growth and reproduction are limited and environments can change drastically. As such, the selective death of adult animals would benefit young and reproductively more privileged populations to facilitate the spreading of genes by young populations under resource-limiting and high-stress conditions. Our work provides an unprecedented example of stress-induced phenoptosis in *C. elegans* and identify a specific transcription factor in a genetic program that likely evolved to promote kin selection during animal evolution. These findings therefore bear broad implications for understanding thermal stress response, programmed organismic death (phenoptosis) and evolutionary biology.

## Materials and methods

### *C. elegans* strains and genetic manipulations

*C. elegans* strains were maintained with standard procedures unless otherwise specified. The N2 Bristol strain was used as the reference wild type, and the polymorphic Hawaiian strain CB4856 was used for genetic linkage mapping and SNP analysis (*Brenner, 1974*; *Davis et al., 2005*). Forward genetic screen for constitutive *asp-17p::GFP* reporter-activating mutants after ethyl methanesulfonate (EMS)-induced random mutagenesis was performed as described previously (*Ma et al., 2012*; *2015*). Single-copy integration of *isy-1p::isy-1::mCherry* transgene was generated using the MosSCI method (*Frøkjaer-Jensen et al., 2008*). To generate *asp-17* null alleles in *C. elegans*, we used CRISPR-Cas9 to induce double-stranded breaks and subsequent non-homologous end joining caused a deletion of *asp-17*. Feeding RNAi was performed as previously described (*Kamath and Ahringer, 2003*). Transgenic strains were generated by germline transformation as described (*Mello et al., 1991*). Transgenic constructs were co-injected (at 10–50 ng/μl) with dominant *unc-54p*::mCherry or *rol-6* markers, and stable extrachromosomal lines of mCherry+ or roller animals were established. Genotypes of strains used are as follows: *daf-16(mu86) I*, *mir-60(n4947) II*; *isy-1 (dma50) V*, *zip-10(ok3462) V*, *asp-17(dma99) V*, *dmaIs10[asp-17p::GFP; unc-54p::mCherry] X*, *dmaIs21[zip-10p::GFP; unc-54p::mCherry]*; *wgIs634[zip-10p::zip-10::EGFP::FLAG + unc-119(+)]*, *oxTi302 [eft-3p::mCherry::tbb-2 3'UTR + Cbr-unc-119(+)]*, *dmaSi1[isy-1p::isy-1::mCherry, unc-119*

(+)], dmaEx95[ges-1p::isy-1(+); rol-6(+)], dmaEx99[isy-1 genomic DNA (2 ng/ul); rol-6(+)], nEx102 [ges-1p::isy-1(+); rol-6(+)], nEx103[ges-1p::isy-1(+); rol-6(+)], dmaEx104[ges-1p::mCherry::3utr(zip-10), rol-6(+)], dmaEx123[hsp-16p::zip-10; rol-6(+)], dmaEx124[hsp-16p::zip-10; rol-6(+)], dmaEx131 [zip-10p::GFP; unc-54p::mCherry].

## Sample and library preparation for RNA sequencing

Control N2 animals and the isy-1 mutants were maintained at 20°C. For cold stress, N2 animals were exposed to 4°C for 2 hrs followed by 1 hr recovery at 20°C. Upon sample collection, the animals were washed down from NGM plates using M9 solution and subjected to RNA extraction using the RNeasy Mini Kit from Qiagen. 1 µg total RNA from each sample was used for sequencing library construction. Each treatment included three biological replicates. The NEBNext rRNA Depletion Kit was used for rRNA depletion. After rRNA depletion, the Agencourt RNAClean XP Beads from Beckman Coulter were used for RNA purification. Then, the NEBNext Ultra Directional RNA Library Prep Kit for Illumina was used for RNA fragmentation, first strand and second strand cDNA synthesis and double-stranded cDNA end repair. Double strand cDNAs were purified using the Agencourt AMPure XP from Beckman Coulter and ligated to adaptors of the NEBNext Multiplex Oligos for Illumina. Finally, the Q5 Hot Start HiFi PCR Master Mix was used for PCR enrichment of the adaptor-ligated DNA. The concentration and quality of the constructed sequencing libraries were measured by using the Agilent High Sensitivity DNA Kit and a Bioanalyzer 2100 from Agilent Technologies. The libraries were submitted to 100 bp paired-end high throughput sequencing using Hiseq-3000 by the Center for Advanced Technology (CAT) of the University of California, San Francisco.

RNA-seq data analysis was performed using a super computer system equipped with multiple processors. The raw reads were trimmed and filtered by the prinseq-lite software (0.20.4) (Schmieder and Edwards, 2011). Reads longer than 30 bp and with a minimum quality score higher than 15 were kept and used for subsequent analyses. The filtered left and right read sets were compared by the Pairfq script to separate paired and single reads. The clean reads were mapped to the C. elegans genome sequence using Hisat2 (2.0.5)(Kim et al., 2015) with default parameters. The number of mapped reads were counted by featureCounts from the Subread package (1.5.0) (Liao et al., 2014). Differential gene expression analysis was performed using the DESeq2 package (Love et al., 2014). Adjusted p-value≤0.05 was used as the threshold to identify the differentially expressed genes. Gene ontology and KEGG pathway enrichment analyses for the differentially expressed genes were conducted using the Cytoscape plugins BiNGO (Maere et al., 2005) and ClueGO (Bindea et al., 2009), respectively. Plots for the mapped reads were generated by IGVtools (Thorvaldsdóttir et al., 2013).

## Quantitative RT-PCR

50 µl pellet animals were resuspended in 250 µl lysis buffer of Quick-RNA MiniPrep kit (Zymo Research, R1055) then lysed by TissueRuptor (Motor unit '8' for 1 min). Total RNA was extracted following the instruction (Zymo Research, R1055). 2 µg RNA/sample was reverse transcribed into cDNA (BioTools, B24408). Real-time PCR was performed by using Roche LightCycler96 (Roche, 05815916001) system and SYBR Green (Thermo Fisher Scientific, FERK1081) as a dsDNA-specific binding dye. qRT-PCR condition was set to 95°C for denaturation, followed by 45 cycles of 10 s at 95°C, 10 s at 60°C, and 20 s at 72°C. Melting curve analysis was performed after the final cycle to examine the specificity of primers in each reaction. Relative mRNA was calculated by ΔΔCT method and normalized to actin. Primers for qRT-PCR: asp-17 (Forward, ATGTTCCGCTGACTGCGAAG; Reverse, TTTCATTCATTTCATCCCAC), F53A9.1, Forward, ACTACGGAAACGGAGGATAC; Reverse, TGGCCGTGATGATGATGATG), srr-6 (Forward, CTCCAAGTCCTGAAGTCGTG; Reverse, GTAGGGA TGGATTGAACTCG), isy-1 (Forward, AGATGCTGAGCGATTCAGAC; Reverse, CTTTCGATAGTCCG TACCAC), zip-10 (Forward, TCGAGATGCTCTTCAACTG; Reverse, CTAACTGCTTGCCGGAG), cpr-3 (Forward, GTAGTGGAGCAGTAACAGGTG; Reverse, CAGTTTGAATTTCGGTGACGG), act-3 (Forward, TCCATCATGAAGTGCGACAT; Reverse, TAGATCCTCCGATCCAGACG).

## Sample preparation and western blot of proteins

Transgenic (isy-1p::isy-1::mCherry and zip-10p::zip-10::EGFP::FLAG) animals were cold shocked (4°C) for 0, 1, 2 or 4 hrs, followed by recovery at 25°C for 1 hr. Animals were harvested and washed three

times with M9 and 20 µl pellet animals were lysed directly in Laemmli Sample Buffer and used for western blot analysis. Proteins were resolved by 15% SDS-PAGE (Bio-Rad, 4561084) and transferred to a nitrocellulose membrane (Bio-Rad, 1620167). Proteins were detected using antibodies against Flag (Sigma, F3165), mCherry (M11217, Life Technologies), Tubulin (Sigma, T5168), H3 (Abcam, ab1791) or HSP90 (Proteintech, 13171–1-AP).

For subcellular fractionation, 50 µl pellet animals were resuspended in 150 µl 1 X cell lysis buffer (Cell Signaling Technology, 9803S) with protease inhibitor cocktail (BioTools, B14002) and 10 µM PMSF, and incubated for 10 min on ice. Animals were lysed by TissueRuptor (Qiagen, 9001271) with Motor unit '6' for 30 s on ice. After incubation on ice for 5 min and centrifugation at 5,000 rpm at 4°C for 2 min, the supernatant was collected as the cytoplasmic part. The nuclear pellet was washed three times with lysis buffer and resuspended in 150 µl RIPA buffer (Thermo Fisher Scientific, P89900) for 30 min on ice, spun at 12,000 rpm for 15 min, and the supernatant was collected as nuclear extract. Tubulin and H3 were separately used as cytoplasm and nuclear loading control. For RNAi experiments, *zip-10p::zip-10::EGFP::FLAG* animals were bleached, and the eggs were laid onto RNAi plates. Animals were harvested as L4/young adults and subject to western blot analysis as described above.

## Chromatin and RNA immunoprecipitation (ChIP-QPCR and RIP-QPCR)

ChIP-QPCR assay was carried out as before with modifications. Briefly, CW-treated animals (4°C for 4 hrs, recovered at 25°C for 1 hr) and control (25°C) animals were harvested and washed by 1 X PBS. The pellet animals were resuspended in cross-linking buffer (1% formaldehyde in 1 X PBS) followed by homogenization using TissueRuptor with Motor unit '4' for 1 min at room temperature. The process was then stopped by addition of glycine (125 mM final concentration). After washing and discarding the supernatant, the pellet was resuspended in lysis buffer and lysed by TissueRuptor with Motor unit '6' for 1 min on ice, with lysate kept on ice for additional 3 min, and then repeated three times. The lysate was centrifuged to collect the supernatant and one percent of the aliquot was used as 'Input'. Lysate was precleared by adding salmon sperm DNA/protein-A agarose beads (Bioworld, 20182011–1), rotating at 4°C for 1 hr. After centrifugation, supernatant was divided equally and added with 50 µg Flag antibody (Sigma, F3165) and mouse IgG (Santa Cruz Biotechnology, sc-2025), respectively. The samples were incubated and rotated overnight at 4°C. Next, salmon sperm DNA/protein-A agarose beads were added for 2 hrs at 4°C. The beads-antibody-TF-DNA complex was washed extensively and the complex and input were diluted with proteinase K buffer. The samples were then incubated at 55°C for 4 hrs and then at 65°C overnight to reverse crosslink. DNA was extracted by phenol-chloroform-isoamylalcohol (Sigma-Aldrich, 77617). *asp-17* promoter was measured by QPCR and calculated by the percent input method. Primers for ChIP-QPCR: *asp-17* promoter (Forward, TTCGCTGCACCTATATGTTG; Reverse, CCGCTAATACCCTTATCAC).

RNA immunoprecipitation (RIP)-QPCR assay was carried out as before with modifications to accommodate our reagents (*Kershner and Kimble, 2010*). Briefly, synchronous day-1 *isy-1p::isy-1::mcherry* animals were divided into two groups. One group is control (25°C) and the other is cold-warming (4°C for 4 hrs, recovered at 25°C for 1 hr). Animals were harvested and washed by M9 buffer until the supernatant was clear, and then washed once in buffer A and twice in lysis buffer. About 250 µl worm pellets were frozen in liquid nitrogen twice and homogenized using TissueRuptor with Motor unit '4' for 1 min on ice. The lysate was kept on ice for 15 min and centrifuged to collect the supernatant and 1% of the aliquot was kept as 'Input'. Equal amount of supernatant was added with RFP-Trap_MA (Chromotek) and rotated for 4 hrs at 4°C. IP magnetic agarose beads were washed and 10% of IP beads were boiled for 6 min in 2X Laemmli Sample Buffer. RNA was eluted from remaining beads using 200 µl lysis buffer of Quick-RNA MiniPrepkit (Zymo Research, R1055) and extracted following the instruction. RNA was quantified with a Nanodrop device. 500 ng RNA was reverse transcribed into cDNA and quantified by the percent input method. Primers for RIP-qPCR: Primary *mir-60* Forward TCGAAAACCGCTTGTTCTTG, Reverse CGATTTCTCAAGTCTTGAAC TAG; *cebp-1* Forward GATCCTTCGCAAGACAAGAC, Reverse CACATTGTCGGTAGGAACGTC.

## Cold tolerance assay

Animals were cultured under non-starved conditions for at least 4 generations at 25°C before cold tolerance assay. For cold tolerance assay of L1-stage animals, bleach-synchronized populations were

kept at 4°C for 96 hrs and then recovered for 4 hrs at 25°C. For cold tolerance assay of adults, animals were raised at 25°C from hatching with excessive bacteria food on agar plates. Well-fed L4 stage animals were transferred to new plates and kept at 25°C overnight to reach day-1 adulthood. To cold shock the animals, agar plates were spread with equal distance on a thin plastic board and transferred to a constant 4°C cold room for 48 hrs or the indicated duration. After cold shock, animals were then moved to 25°C for recovery for 4 hrs before scoring survival rates. Animals were scored as dead if they showed no pumping and movement upon light touch with the body necrosis subsequently confirmed.

## Imaging and fluorescence quantification

smFISH of *C. elegans* and imaging were performed as previously described (*Ji and Oudenaarden, 2005*). For fluorescence imaging, spinning-disc confocal and digital automated epifluorescence microscopes (EVOS, Life Technologies) were used to capture images of animals after RNAi or CW treatments. Synchronous population of worms were randomly picked and treated with 1 mM levamisole water solution to paralyze the animals. The animals were mounted on an agar pad on a slide and aligned for imaging. Identical conditions and settings were used for both control and test groups. For quantification of fluorescence images, the animals in the images were outlined and signals were quantified by ImageJ software. The intensity of an individual animal was obtained by dividing the total signal by the area of that animal. The average intensity of the control group was set to be 1.0, to which all other intensities were normalized. Graphpad Prism software was used to plot the data.

## Small RNA-seq and bioinformatics

For small RNA sequencing, total RNA was isolated by the Quick-RNA MiniPrep kit (Zymo Research, R1055) that yields total RNA including small RNAs ranging 17–200 nt. RNA samples extracted from triplicate N2 animals and *isy-1* mutants were submitted to Beijing Genomics Institute for small RNA library construction and sequencing. The low-quality reads were filtered and clean reads were mapped to the *C. elegans* genome using Bowtie2 program (*Langmead and Salzberg, 2012*). MiR-Deep2 (*Friedländer et al., 2012*) was used to characterize known and predict novel miRNAs. The small RNA expression level was calculated as TPM (transcript per million). Differentially expressed small RNAs were detected by DESeq2 (*Love et al., 2014*). The threshold for differentially expressed sRNAs was adjusted p-value≤0.05 and the absolute value of Log2ratio ≥1. Targets of miRNAs were predicted by TargetScan (*Jan et al., 2011*), RNAhybrid (*Krüger and Rehmsmeier, 2006*) and miRanda (*John et al., 2004*) using default parameters.

## Statistical analysis

Data were analyzed using GraphPad Prism Software (Graphpad, San Diego, CA) and presented as means ± S.D. unless otherwise specified with p values calculated by unpaired Student's t-tests, one-way or two-way ANOVA (comparisons across more than two groups) and adjusted with Bonferroni's corrections.

## Acknowledgements

We thank the *Caenorhabditis* Genetics Center, National BioResource Project in Japan and the Million Mutation Project for *C. elegans* strains. The work was supported by NIH grants R01GM117461, R00HL116654, ADA grant 1–16-IBS-197, Pew Scholar Award, Alfred P Sloan Foundation Fellowship, and Packard Fellowship in Science and Engineering (DKM), R01AG032435 (YW) and a CPSF postdoctoral fellowship (WJ).

## Additional information

### Funding

| Funder | Grant reference number | Author |
| --- | --- | --- |
| China Postdoctoral Science Foundation | Post-doctoral fellowship | Wei Jiang |

| | | |
|---|---|---|
| National Institute on Aging | R01AG032435 | Yuehua Wei |
| National Institute of General Medical Sciences | R01GM117461 | Dengke K Ma |
| Pew Charitable Trusts | Pew Scholar Award | Dengke K Ma |
| American Diabetes Association | 1–16-IBS-197 | Dengke K Ma |
| Alfred P. Sloan Foundation | | Dengke K Ma |
| David and Lucile Packard Foundation | Fellowship in Science and Engineering | Dengke K Ma |
| Esther A. and Joseph Klingenstein Fund | | Dengke K Ma |
| National Heart, Lung, and Blood Institute | R00HL116654 | Dengke K Ma |

The funders had no role in study design, data collection and interpretation, or the decision to submit the work for publication.

## Author contributions

Wei Jiang, Data curation, Formal analysis, Validation, Investigation, Visualization, Methodology, Writing—original draft, Writing—review and editing; Yuehua Wei, Yong Long, Data curation, Formal analysis, Validation, Investigation, Visualization, Methodology, Writing—original draft; Arthur Owen, Investigation, Methodology, Writing—review and editing; Bingying Wang, Resources, Investigation, Methodology; Xuebing Wu, Data curation, Formal analysis, Investigation, Visualization; Shuo Luo, Investigation, Methodology; Yongjun Dang, Resources, Methodology; Dengke K Ma, Conceptualization, Resources, Data curation, Formal analysis, Supervision, Funding acquisition, Validation, Investigation, Visualization, Methodology, Writing—original draft, Project administration, Writing—review and editing

## Author ORCIDs

Wei Jiang http://orcid.org/0000-0001-7615-0900
Xuebing Wu http://orcid.org/0000-0003-0369-5269
Yongjun Dang http://orcid.org/0000-0001-7237-1132
Dengke K Ma http://orcid.org/0000-0002-5619-7485

## Decision letter and Author response

Decision letter https://doi.org/10.7554/eLife.35037.025
Author response https://doi.org/10.7554/eLife.35037.026

## Additional files

### Supplementary files

• Supplementary file 1. A specific subset of CW-inducible genes is dependent on ZIP-10. Shown is a table of fold induction for gene expression levels determined by QPCR measurements of top-ranked randomly selected CW-inducible genes in wild type and *zip-10* mutant animals.
DOI: https://doi.org/10.7554/eLife.35037.018
• Transparent reporting form
DOI: https://doi.org/10.7554/eLife.35037.019

### Major datasets

The following datasets were generated:

| Author(s) | Year | Dataset title | Dataset URL | Database, license, and accessibility information |
|---|---|---|---|---|
| Wei J, Yuehua W, Yong L, Arthur O, Bingying W, Xuebing W, Shuo L, Yongjun D, Dengke KM | 2018 | CW RNA seq | https://www.ncbi.nlm.nih.gov/bioproject/?term=PRJNA430003 | Publicly available at NCBI BioProject (Accession no. PRJNA430003) |
| Wei J, Yuehua W, Yong L, Arthur O, Bingying W, Xuebing W, Shuo L, Yongjun D, Dengke KM | 2018 | Small RNA seq in isy-1 mutants | https://www.ncbi.nlm.nih.gov/bioproject/?term=PRJNA430140 | Publicly available at NCBI BioProject (Accession no. PRJNA430140) |

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
