## [Decision Letter]

[Editors’ note: a previous version of this study was rejected after peer review, but the authors submitted for reconsideration. The first decision letter after peer review is shown below.]

Thank you for submitting your work entitled "A Genetic Program Underlies Cold-warming Response and Promotes Phenoptosis" for consideration by *eLife*. Your article has been reviewed by three peer reviewers, and the evaluation has been overseen by a Reviewing Editor (Julie Ahringer) and a Senior Editor (Detlef Weigel). The following individuals involved in review of your submission have agreed to reveal their identity: Shouhong Guang (Reviewer #1); Meng C Wang (Reviewer #2); Ding Xue (Reviewer #3.

Our decision has been reached after consultation between the reviewers. Based on these discussions and the individual reviews below, we regret to inform you that your work, at least in its current form, will not be considered further for publication in *eLife*.

We appreciate both the general interest in the phenomenon you describe and the strengths of several of your experiments, with which you have made progress in understanding hypothermic stress and the cold-warming (CW) response, an important but little studied topic. At the same time, there was broad agreement that the study is not yet at a sufficient level for a general audience. In particular, the precise mechanism of *isy-1* action is still unclear. Similar, it is not yet known whether the genes identified are specific to CW or alternatively act in other stress pathways. Additionally, an important paper by Kato et al., 2016 linking mir-61 to regulation of *zip-10* and *asp-17* appears to have been missed; work that is relevant to the mechanism of *isy-1* and the specificity of the genes studied here.

Our criticisms notwithstanding, we are very interested in the CW phenomenon. If you decide, based on the reviews, not to go immediately elsewhere with the work for publication, but rather to add more mechanistic detail before resubmitting it, we would encourage you to consider *eLife* again for such a future submission. If the reviewers' comments are appropriately addressed, we would very likely review such a new submission again and would try to consult the same reviewers. In the meantime, we hope that you find the reviewers' specific comments helpful.

Essential revisions:

Reviewer #1:

Altogether an interesting paper, with good data. It touches on important points, and I think it would be valuable to the community and would be relevant to anyone who is interested in hypothermic stress regulation. But some revisions must be made. Context is frequently omitted making this paper difficult to read. A longer format may be warranted to accommodate additional context and background.

1) *asp-17* was chosen as a robust CW-inducible reporter gene to search for regulators and identified *isy-1/zip-10* pathway. How general is this *isy-1/zip-10* regulation? How many other genes use this mechanism to respond to CW induction and how many don't? In Figure 3I, F39A9.1 is responsive to CW but independent of *zip-10*, what is the mechanism of this regulation? And *srr-6* exhibits a more pronounced upregulation to CW in *zip-10* KO mutant.

2) Figure 4C, does *cpr-3* contains similar ZIP-10-binding motif? A genome wide ChIP-seq of ZIP-10 with and without CW stress, and in *isy-1(dma50*) mutants will greatly increase the paper.

3) The Figure 4—figure supplement 1A-1F of microRNA deep sequencing is not involved in this paper at all, which can either be taken out or need more elaboration. Meanwhile, in the methods section, the deep-sequencing of small RNAs need elaboration. For example, did they use 5'-phosphate-dependent or independent method for sequencing? A recent paper showed that CW-induction (very similar to the treatment in this paper) elicits risiRNA expression, has this category of small RNAs been analyzed.

4) The Discussion section is very hard to read, both the length of sentences and the logic.

5) It is very interesting how CW induction upregulates *zip-10* expression. Is there any molecular reasoning of this regulation? In the genetic screening, has the author tried to isolate mutants that failed to respond to CW stress induction, in addition to the constitutive *asp-17::GFP* expression mutants?

6) In Figure 1—figure supplement 1C, in unc-43 mutant, the expression of *asp-17* is also decreased with and without CW stress. How about the relative change? And in TRPA-1 mutant, the upregulation of *asp-17* is actually more pronounced. CW likely involves calcium signaling is an interesting observation, which may need further discussion or investigation.

7) In Figure 4B, there is a very good correlation between *isy-1*/control vs. CW/control. Does this mean CW actually directly regulates *isy-1*?

Reviewer #2:

In this manuscript, the authors have used *C. elegans* as an animal model to investigate the molecular mechanisms that organisms utilize to handle cold shock and sequent warm-up (cold-warm stress response). From candidate genes upregulated by cold-warm stress, the authors have identified *asp-17*, an aspartyl-like protease and generated its GFP reporters for an EMS mutagenesis screen. From the screen, the authors discovered an *isy-1* mutant that leads to constitutive induction of *asp-17* without cold-warm stress. *isy-1* encodes a conserved RNA-binding protein and acts in intestinal cells to regulate *asp-17* expression cell-autonomously. Furthermore, they discovered that the ZIP-10 transcription factor functions downstream of *isy-1* and directly regulates *asp-17* expression. Inactivation of *zip-10* or *asp-17* increases organism survival upon cold-warm stress, while inactivation of *isy-1* also increases organism survival upon cold-warm stress. Overall, the manuscript addresses an interesting question that remains poorly understood and is well written.

1) In this study, the authors have used cold-shock and sequent warm-up as an experimental system to assay a cold-warm stress response. In this experimental set up, both phases, cold and warm, are important, and can have distinct molecular and physiological changes related to organism survival. In fact, as shown in Figure 3E and 3G, the protein levels of ZIP-10 show very different dynamics with different duration of cold shock and warm-up. It seems that ZIP-10 levels won't be induced without warm-up. Is this true? So which phase does *zip-10* play a crucial role in regulating organism responses and survival? How about *asp-17*? The authors should consider characterizing the dynamic changes of *zip-10* and *asp-17* during cold shock and during warm-up.

Also, the induction of *zip-10* goes away with increasing warming time. Is *isy-1* responsible for regulating this transient induction? In the *isy-1* inactivated condition, will *zip-10* levels stay on with increasing warming time?

2) *asp-17* is induced by the *isy-1* mutant without cold-warm stress, and this induction can be further enhanced by cold-warm stress (similar induction folds in both conditions, Figure 3A). It is likely that *isy-1* just controls the *asp-17* expression level in general. Then how about *zip-10*? Does *isy-1* just control its levels in general, or more specific to cold-warm stress?

3) *zip-10* is a transcription factor. The authors showed that it is transcriptionally induced upon cold-warm stress and increases its occupancy at the promoter of *asp-17*. Does this transcription factor regulate *asp-17* cell-autonomously in intestinal cells? Where is this transcription factor localized in the cell, cytosol or nucleus? Does cold-warm stress affect its cellular localization? From the WB results shown in Figure 3—figure supplement 2, it is hard to tell. The authors should consider conducting immunostaining.

4) The authors showed that the *isy-1* mutant is more resistant to cold-warm stress, which can be partially suppressed by *daf-16*. Does *daf-16* affect the tolerance of the *zip-10* mutant? Also, does HSF-1 affect the tolerance of the *isy-1* mutant?

5) When comparing Figures 4E and 4F, it seems like that heat-shock pre-treatment increases tolerance to cold-warm stress (0% death rate vs. 65% death rate). Is this true? If yes, could this be the reason why the *isy-1* mutant is more resistant, since hsp-16.2 and hsp-16.41 are induced in the *isy-1* mutant.

Reviewer #3:

Hypothermia is a type of stress that might induce phenoptosis in severe conditions. This paper identifies a molecular pathway that regulates the cold-warming response in *C. elegans*. Following the "cold-warming" (CW) treatment, the authors did an RNA-Seq analysis to identify genes that are differentially expressed with or without CM. They identified *asp-17*, an aspartyl protease that was upregulated following CW treatment. To identify genes that regulate the expression of *asp-17*, they did an EMS mutagenesis and isolated a mutation in the *isy-1* gene that constitutively up-regulates the transcriptional expression of *asp-17* without CW treatment. To identify transcription factors that might be regulated by *isy-1*, they performed another RNA-Seq analysis using WT and *isy-1* mutants and identified *zip-10*. Further experiments showed that *isy-1* suppresses *zip-10* expression, whereas CW upregulates *zip-10* expression. They also suggest that *zip-10* has a pro-death role important for kin selection in evolution. Their studies suggest that CW induces *zip-10* upregulation and elevated death in older adults compared with larvae, thus supporting the theory of kin selection.

This is a strong paper that reveals critical components involved in a poorly understood stress-response process. The experimental approach is straightforward, and the data are solid. The conclusions are supported by multiple independent experiments. The main drawback is the lack of mechanistic understanding of how ISY-1 regulates ZIP-10. I support the publication of this paper in *eLife* with appropriate revisions.

1) Figure 1B, align "Ctr" or "CW" labels with the data. Figure 1F: In the legends they mentioned 'signals indicated by red', which should be yellow.

2) Figure 4D, to show ZIP-10 binds directly to the *asp-17* promoter, in vitro DNA binding assays need to be performed and ZIP-10 binding sites should be deleted as a negative control.

3) CPR-4 inhibits germ cell death, instead of promoting germ cell death (Discussion section).

4) Since CPR-3 is one of the ZIP-10 downstream targets and is another protease like ASP-17, the consequence of CPR-3 overexpression should be examined.

Reviewing Editor:

The authors address hypothermic stress and the cold-warming (CW) response, an interesting but little studied topic. They have made progress by identifying a cascade of genes that regulate or are regulated by CW. This is a good start, but the mechanism of *isy-1* action is still unknown and the specificity of the genes in CW is wasn't determined.

The authors have missed an important paper that has linked *mir-60* to regulation of *asp-17* and *zip-10* (Kato et al., 2016). Both genes are upregulated in *mir-60* mutants and *zip-10* is a strong candidate for being a direct target of *mir-60* (the mRNA has several good predicted binding sites). Given that human ISY1 is involved in microRNA processing, a plausible model is that *isy-1* mutants reduce expression of *mir-60* (which is intestine specific), which leads to upregulation of ZIP-10, which then induces *asp-17* expression. The authors need to take this previous work into account, both by discussing their results in light of this paper and by directly investigating the potential link with *mir-60*.

If ISY-1 is involved in microRNA processing, then it is probably not specifically involved in CW. Instead it may non-specifically affect this process through processing of microRNAs. If so, loss of other factors that process microRNAs (e.g. dicer, drosha) would be expected to activate the *asp-17* reporter. This and whether loss of *isy-1* affects microRNA processing should be tested. Interestingly, Figure 4—figure supplement 1 shows that *mir-60* is downregulated in *isy-1* mutants.

ZIP-10 has been previously linked to the innate immunity pathway, and its expression is upregulated upon exposure to bacterial pathogens (Shapira et al., 2006). BATF3, the human ortholog of ZIP-10, has also been linked to immune system regulation and response to pathogens. Is *asp-17* upregulated upon exposure to pathogens?

Liang et al., 2007 (not cited) showed that *zip-10* is expressed in the intestine.

What was the rational for focusing on *zip-10*? Other TFs are also upregulated in *isy-1* mutants (*jun-1, cebp-1, fos-1, zip-2*, and others). Interestingly, *zip-2* mediates response to *Pseudomonas aeruginosa* infection (Estes et al., 2010).

The authors propose that ZIP-10 may promote organismal death. Could the authors test if it has such a role in pathogen infection?

[Editors’ note: what now follows is the decision letter after the authors submitted for further consideration.]

Thank you for resubmitting your work entitled "A Genetic Program Mediates Cold-warming Response and Promotes Phenoptosis in *C. elegans*" for further consideration at *eLife*. Your revised article has been favorably evaluated by Detlef Weigel (Senior editor), Julie Ahringer (a Reviewing editor) and three reviewers (Meng Wang, Shouhong Guang, and Ding Xue).

There was agreement that your revised manuscript has been significantly improved and largely addresses previous concerns and comments. In particular, more mechanistic insight was provided regarding how ISY-1 regulates ZIP-10 (through *mir-60*), although how *mir-60* negatively regulates *zip-10* remains unclear. A few concerns remain that should be addressed:

1) CW and ISY-1 regulate *zip-10* predominantly at the transcriptional level. Because microRNAs can function in the cytoplasm to downregulate targeted mRNAs or inhibit translation, *mir-60* may not directly prohibit the transcription of *zip-10*. It would be of interest to discuss this potential for post-transcriptional regulation in models of of ISY-1and *mir-60* may regulate *zip-10*.

2) In the new Figures 4B-4C, have the numbers been normalized to the amount of immunoprecipitated proteins?

3) The legend of panels D & E are missing for Figure 3—figure supplement 1.

4) In Figure 4—figure supplement 1, experimental details and statistics are missing. Including the numbers of fold change in *zip-10* mutant would be helpful.

5) For western-blotting of the proteins, the work used mixed-staged animals. Do ISY-1 and ZIP-10 exhibit stage-specific expression? A population of mixed-staged animals may contain different combination of the stages, which can complicate the explanation.

6) In the Materials and methods section, the RNeasy Mini Kit from Qiagen was used to purify total RNAs, which are usually longer than 200 nt. How small RNAs were isolated for library construction and deep sequencing needs clarification. Endogenous siRNAs were missed from the Figure 5—figure supplement 1, suggesting that the 5'-dependent method was used to sequence the small RNA library.

7) In Figure 4E, please use different colors to represent different bars. The p-values between *zip-10* bars need to be shown. In addition, the relative mRNA level of *zip-10* in *zip-10(ok3462*) is zero and the relative mRNA level of *zip-10* in the *mir-60; zip-10* double is around 1.5. These results are not consistent. Please explain.

8) Results section, third paragraph, Figure 2A should be Figure 2B.

---

## [Author Response]

[Editors’ note: the author responses to the first round of peer review follow.]

Reviewer #1:Altogether an interesting paper, with good data. It touches on important points, and I think it would be valuable to the community and would be relevant to anyone who is interested in hypothermic stress regulation. But some revisions must be made. Context is frequently omitted making this paper difficult to read. A longer format may be warranted to accommodate additional context and background.

Thank you for the suggestion. We have resorted to a longer Research Article format, which we agree is more appropriate for accommodating additional context, background and new results needed to improve readability and address the comments and questions raised by the reviewers and editors.

1) asp-17 was chosen as a robust CW-inducible reporter gene to search for regulators and identified isy-1/zip-10 pathway. How general is this isy-1/zip-10 regulation? How many other genes use this mechanism to respond to CW induction and how many don't? In Figure 3I, F39A9.1 is responsive to CW but independent of zip-10, what is the mechanism of this regulation? And srr-6 exhibits a more pronounced upregulation to CW in zip-10 KO mutant.

We agree this is an important question to address. To assess specificity of gene regulation mediated by ISY-1/ZIP-10, we have performed QPCR quantification of the expression levels of 17 genes (in addition to the 4 reported in the initial submission) randomly selected from the gene set commonly regulated by ISY-1 and CW. The results indicate striking specificity: only *asp-17, cpr-3* and *ZK896.4* strictly depend on ZIP-10 while the remaining set of genes are up-regulated by CW even in *zip-10* mutants (new Figure 4—figure supplement 1). This indicates that ZIP-10 likely promotes a dedicated genetic program in response to CW, consistent with the induction of ZIP-10 specifically by CW stress (Figure 3—figure supplement 2B) and its mutant phenotype in CW adaptation (new Figure 5). Although identifying an exhaustive list of stringent ZIP-10 target genes will require additional RNAseq and tissue-specific ChIP experiments, we believe *asp-17* is the major and physiologically relevant target since LOF phenotype of *zip-10* is largely recapitulated by that of *asp-17* (new Figure 5E). Investigating mechanisms of regulation of *F39A9.1* and *srr-6* is underway and the results will be described elsewhere given the focus/scope of this paper and the ZIP-10 independence of their regulation by CW.

2) Figure 4C, does cpr-3 contains similar ZIP-10-binding motif? A genome wide ChIP-seq of ZIP-10 with and without CW stress, and in isy-1(dma50) mutants will greatly increase the paper.

Although the *cpr-3* promoter does not contain a canonical ZIP-10-binding AT-rich motif as predicted by the stringent criteria of the program (new Figure 5C), we manually checked the promoter sequence and note that it does contain conserved AT-rich motifs resembling the canonical ZIP-10 binding motif (e.g. agaatttttttaaattttcaacaaaa, at -441 nt of its transcriptional start site) that might allow binding of ZIP-10 at lower affinity or in combination with other contextual transcriptional binding partners. Since ZIP-10 seems to be exclusively present in intestine, whole-organism ChIP-seq of ZIP-10 would be limited in identifying additional relevant targets. We would like to focus on *asp-17*, the validated transcriptional target of ZIP-10 that confers major causal effects in CW responses (new Figure 5).

3) The Figure 4—figure supplement 1A-1F of microRNA deep sequencing is not involved in this paper at all, which can either be taken out or need more elaboration. Meanwhile, in the methods section, the deep-sequencing of small RNAs need elaboration. For example, did they use 5'-phosphate-dependent or independent method for sequencing? A recent paper showed that CW-induction (very similar to the treatment in this paper) elicits risiRNA expression, has this category of small RNAs been analyzed.

Thank you for the suggestion. Given the note from the reviewing editor that *mir-60* is downregulated in *isy-1* mutants, we have confirmed the finding by QPCR and performed functional experiments to test causal involvement of *mir-60*. As suggested by the reviewing editor, we have explored the mechanistic connection of ISY-1 to *mir-60* and found indeed ISY-1 binds directly to the primary transcript of *mir-60*, based on RNA immune-precipitation of mCherry-tagged ISY-1, followed by QPCR quantification (new Figure 4). Furthermore, *mir-60* mutants exhibited elevated expression of *asp-17* in a manner that requires *zip-10*, thus linking ISY-1 to ZIP-10 mechanism of regulation. Therefore, we’d like to keep the microRNA profiling result including that of *mir-60*, which is now more directly relevant for this paper.

We did not distinguish the 5'-phosphate-dependent or independent methods or analyze risiRNA expression. Nonetheless, we performed new RNA immune-precipitation experiments and found that levels of *mir-60* bound to ISY-1 were slightly but significantly increased by CW, consistent with enhanced processing of small RNA by ISY-1, e.g. *mir-60* in particular as a feedback mechanism to curb overexpression and toxicity of ZIP-10 after CW. Although potential link from ISY-1 to risiRNA was not explored, the interesting result of CW regulation of risiRNA indicates a perhaps similar mechanism and function in organismic homeostatic control and now included in the paper for brief discussion.

4) The Discussion section is very hard to read, both the length of sentences and the logic.

We have reorganized and rewritten the Discussion section to improve readability.

5) It is very interesting how CW induction upregulates zip-10 expression. Is there any molecular reasoning of this regulation? In the genetic screening, has the author tried to isolate mutants that failed to respond to CW stress induction, in addition to the constitutive asp-17::GFP expression mutants?

Thank you for the comment. The up-regulation of *zip-10* is indeed striking and very interesting from molecular mechanistic points of view. Our results indicate that *zip-10* up-regulation by CW is largely mediated by transcriptional mechanisms, since GFP driven by the *zip-10* promoter (*zip-10p::GFP*) was up-regulated by CW while its 3’utr was not (new Figure 4F). We have used EMS mutagenesis to isolate viable mutants with *zip-10p::GFP* that fail to respond to CW induction but unfortunately the penetrance of such mutants were generally too low (~5-20%) to be tractable for genetic mapping and molecular cloning. We reason that the upstream stress-responding regulators might be essential (precedents include genes e.g. *skn-1*) or genetically redundant (precedents include GPCRs) so that positive gene regulators of CW responses are harder to identify than negative gene regulators like *isy-1*.

6) In Figure 1—figure supplement 1C, in unc-43 mutant, the expression of asp-17 is also decreased with and without CW stress. How about the relative change? And in TRPA-1 mutant, the upregulation of asp-17 is actually more pronounced. CW likely involves calcium signaling is an interesting observation, which may need further discussion or investigation.

We agree with this reviewer that the baseline level of *asp-17* is already low in *unc-43* mutants and involvement of calcium signaling based on *unc-43* and *trpa-1* results alone is difficult to interpret. The relative change is not significant compared with wild type. Thus, we have removed this result in the revised manuscript to avoid any confusion.

7) In Figure 4B, there is a very good correlation between isy-1/control vs. CW/control. Does this mean CW actually directly regulates isy-1?

Thank you for pointing out this observation to clarify. Our results clearly indicate that CW does not directly regulate ISY-1 since endogenous *isy-1* mRNA and mCherry-tagged proteins are not changed by CW (Figure 2—figure supplement 2). In our model, *zip-10* up-regulation by CW and ISY-1 occurs likely in parallel pathways and is mediated largely by transcriptional mechanisms (based on *zip-10p::GFP* up-regulation). We clarified in the revised paper that marked correlation between *isy-1*/control vs. CW/control strongly supports common genetic regulation of additional unidentified transcriptional regulators that control *zip-10* rather than direct causal regulation of *isy-1* by CW.

Reviewer #2:[…] 1) In this study, the authors have used cold-shock and sequent warm-up as an experimental system to assay a cold-warm stress response. In this experimental set up, both phases, cold and warm, are important, and can have distinct molecular and physiological changes related to organism survival. In fact, as shown in Figure 3E and 3G, the protein levels of ZIP-10 show very different dynamics with different duration of cold shock and warm-up. It seems that ZIP-10 levels won't be induced without warm-up. Is this true? So which phase does zip-10 play a crucial role in regulating organism responses and survival? How about asp-17? The authors should consider characterizing the dynamic changes of zip-10 and asp-17 during cold shock and during warm-up.

Thank you for the comments and raising these important questions for us to clarify. Yes, it is true that neither *asp-17* nor *zip-10* levels are induced unless with warm-up (Figure 1E and new Figure 4G). Our data indicate that their induction require both cold and warm phases but it is during the warm phase that the genetic program comes to play in regulating organismic response and survival. This is also consistent with the notion that very severe hypothermic stress actually arrests most of cellular activities, as discussed in the Introduction in the context of rationale why we chose to use cold-warming rather than simply cold shock as the stress paradigm. In the revised manuscript, we have clarified this issue by additional discussion and also provided additional experimental characterization of the dynamic ranges of *zip-10* and *asp-17* regulation during CW (please see Figure 1E, Figure 3G and Figure 4G).

Also, the induction of zip-10 goes away with increasing warming time. Is isy-1 responsible for regulating this transient induction? In the isy-1 inactivated condition, will zip-10 levels stay on with increasing warming time?

We performed new RNA immune-precipitation experiments and found that the *zip-10*-regulating *mir-60* binds to ISY-1 directly and levels of *mir-60* bound to ISY-1 were slightly but significantly increased by CW (new Figure 4A-4D). ZIP-10 is transiently induced by CW, has constitutively high baseline level and is further induced by CW in *isy-1* mutants, indicating that *isy-1* gates *zip-10* expression via *mir-60*, while not responsible for triggering the transient induction of *zip-10* by CW. We also performed new experiments to examine effects of increasing warming time on *zip-10* induction (new Figure 4G). As predicted, *isy-1* RNAi-treated animals exhibit relatively high ZIP-10 levels under baseline conditions and even more pronounced ZIP-10 up-regulation upon CW. However, ZIP-10 levels did not stay on over extended increasing warming time likely because of residual *mir-60* functions as reasoned above and/or additional post-translational mechanisms promoting degradation of ZIP-10.

2) asp-17 is induced by the isy-1 mutant without cold-warm stress, and this induction can be further enhanced by cold-warm stress (similar induction folds in both conditions, Figure 3A). It is likely that isy-1 just controls the asp-17 expression level in general. Then how about zip-10? Does isy-1 just control its levels in general, or more specific to cold-warm stress?

Please see the above response to Point 1. Our previous and new data strongly indicate that CW but not other stress including hypoxia up-regulates *zip-10* transcription, leading to up-regulation of *asp-17* transcription, whereas ISY-1 regulates *zip-10* (and thereby *asp-17*) via binding and processing of *mir- 60*, thus gating (permissive) but not triggering (instructive) the induction of *zip-10* in response to CW.

3) zip-10 is a transcription factor. The authors showed that it is transcriptionally induced upon cold-warm stress and increases its occupancy at the promoter of asp-17. Does this transcription factor regulate asp-17 cell-autonomously in intestinal cells? Where is this transcription factor localized in the cell, cytosol or nucleus? Does cold-warm stress affect its cellular localization? From the WB results shown in Figure 3—figure supplement 2, it is hard to tell. The authors should consider conducting immunostaining.

Like *asp-17, zip-10* is exclusively expressed in intestine, based on our promoter-driven GFP reporters (Figure 3—figure supplement 2A) and single cell RNAseq data recently published by the Waterston lab (Cao et al., 2017), supporting cell-autonomous regulation. We have attempted to generate several arrays of ectopically over-expressed ZIP-10::GFP fusion constructs at varying concentrations to visualize its subcellular localizations but were not able to obtain viable animals, likely because of its toxicity, consistent with a role in promoting phenoptosis. Immunostaining of the low-copy FLAG-tagged ZIP-10 in single animals did not yield sufficiently strong specific signals for resolving cellular localization. However, we were able to use Western blot to obtain evidence of ZIP-10 abundance change and nuclear accumulation because of increased animal numbers (>100/per blot) for the Western blot assay.

4) The authors showed that the isy-1 mutant is more resistant to cold-warm stress, which can be partially suppressed by daf-16. Does daf-16 affect the tolerance of the zip-10 mutant? Also, does HSF-1 affect the tolerance of the isy-1 mutant?

We performed new experiments and statistic calculation indicates that *daf-16* slightly decreased the tolerance of *zip-10* mutants (p = 0.0464, unpaired t-test, n=3 independent assays for N>30 animals for each group) whereas *hsf-1* more pronouncedly decreased the tolerance of *isy-1* mutants (p = 0.0128, unpaired t-test, n=3 independent assays for N>30 animals for each group). These results suggest that ZIP-10 promotes phenoptosis in response to CW partially by antagonizing DAF-16 effects and that ISY-1 regulates broad programs of stress response including the HSF-1 pathway, consistent with pleiotropic effects of *isy-1* mutants as indicated in the Model and discussed in the text.

5) When comparing Figures 4E and 4F, it seems like that heat-shock pre-treatment increases tolerance to cold-warm stress (0% death rate vs. 65% death rate). Is this true? If yes, could this be the reason why the isy-1 mutant is more resistant, since hsp-16.2 and hsp-16.41 are induced in the isy-1 mutant.

We apologize for the confusion due to inadequacy of our description – Figure 4F is based on organismic death rate under non-CW stressed conditions to assay effects of *zip-10* overexpression. We have now further clarified this in the Figure legend. Nonetheless, we agree with this reviewer’s reasoning that *isy-1* mutants are more resistant to CW stress likely because of baseline induction of many DAF-16 and HSF-1 targets that can help cope with stress, as now discussed in the text.

Reviewer #3:Hypothermia is a type of stress that might induce phenoptosis in severe conditions. This paper identifies a molecular pathway that regulates the cold-warming response in C. elegans. Following the "cold-warming" (CW) treatment, the authors did an RNA-Seq analysis to identify genes that are differentially expressed with or without CM. They identified asp-17, an aspartyl protease that was upregulated following CW treatment. To identify genes that regulate the expression of asp-17, they did an EMS mutagenesis and isolated a mutation in the isy-1 gene that constitutively up-regulates the transcriptional expression of asp-17 without CW treatment. To identify transcription factors that might be regulated by isy-1, they performed another RNA-Seq analysis using WT and isy-1 mutants and identified zip-10. Further experiments showed that isy-1 suppresses zip-10 expression, whereas CW upregulates zip-10 expression. They also suggest that zip-10 has a pro-death role important for kin selection in evolution. Their studies suggest that CW induces zip-10 upregulation and elevated death in older adults compared with larvae, thus supporting the theory of kin selection.This is a strong paper that reveals critical components involved in a poorly understood stress-response process. The experimental approach is straightforward, and the data are solid. The conclusions are supported by multiple independent experiments. The main drawback is the lack of mechanistic understanding of how ISY-1 regulates ZIP-10. I support the publication of this paper in eLife with appropriate revisions.

Thank you for the comments. Regarding mechanistic understanding of how ISY-1 regulates ZIP-10, we have performed new experiments and explored the connection of ISY-1 to mir-60, as suggested by the reviewing editor, and found indeed ISY-1 binds directly to the primary transcript of mir-60, based on RNA immune-precipitation of mCherry-tagged ISY-1, followed by QPCR quantification. Furthermore, mir-60 mutants exhibited elevated expression of zip-10 (and thereby asp-17) and reduced cold tolerance (new Figure 4A-4G and Figure 5H), thus linking ISY-1 to ZIP-10 mechanism of regulation. Human ISY1 has been shown to be important for processing of certain primary microRNA transcripts (Du et al., 2015). We suggest that ISY1’s role in microRNA processing is conserved in metazoans and identify a specific microRNA target that is important for organismic response to CW.

1) Figure 1B, align "Ctr" or "CW" labels with the data. Figure 1F: In the legends they mentioned 'signals indicated by red', which should be yellow.

Thank you. We have corrected it.

2) Figure 4D, to show ZIP-10 binds directly to the asp-17 promoter, in vitro DNA binding assays need to be performed and ZIP-10 binding sites should be deleted as a negative control.

We agree in vitro DNA binding would support our in vivo ChIP results. Given the 16 predicted highconfidence ZIP-10 binding motifs in the *asp-17* promoter, there is likely redundancy of ZIP-10 binding and their comprehensive dissection will be substantial work beyond the scope of this study. Given the strong genetic causality evidence and correlative ChIP results, we conclude that ZIP-10 is required for *asp-17* induction by CW but have toned down the specific roles of the predicted ZIP-10 binding motifs.

3) CPR-4 inhibits germ cell death, instead of promoting germ cell death (Discussion section).

Thank you. We have corrected it and modified the discussion.

4) Since CPR-3 is one of the ZIP-10 downstream targets and is another protease like ASP-17, the consequence of CPR-3 overexpression should be examined.

Thank you for the suggestion. We have performed the experiment and found that ectopic *cpr-3* overexpression also led to enhanced death rates as *asp-17* did, supporting a phenoptosis program activated by *zip-10* (p < 0.001, unpaired t-test, n=3 independent assays for N>30 animals for each group). However, unlike viable *asp-17* null allele mutant, deletion of *cpr-3* by *ok1788* homozygotes caused mildly Unc and mid-larval arrest phenotype, precluding us from assessing its physiological role in cold tolerance. As such, we focus on analysis of *asp-17* rather than *cpr-3* in the current study.

Reviewing Editor:The authors address hypothermic stress and the cold-warming (CW) response, an interesting but little studied topic. They have made progress by identifying a cascade of genes that regulate or are regulated by CW. This is a good start, but the mechanism of isy-1 action is still unknown and the specificity of the genes in CW is wasn't determined.The authors have missed an important paper that has linked mir-60 to regulation of asp-17 and zip-10 (Kato et al., 2016). Both genes are upregulated in mir-60 mutants and zip-10 is a strong candidate for being a direct target of mir-60 (the mRNA has several good predicted binding sites). Given that human ISY1 is involved in microRNA processing, a plausible model is that isy-1 mutants reduce expression of mir-60 (which is intestine specific), which leads to upregulation of ZIP-10, which then induces asp-17 expression. The authors need to take this previous work into account, both by discussing their results in light of this paper and by directly investigating the potential link with mir-60.

Thank you for the comments. The paper by Kato et al., 2016 inadvertently omitted in our initial submission (thank you for bringing this to our attention) proved informative in suggesting a candidate ISY-1 target microRNA *mir-60*. We have since performed new experiments and explored the connection of ISY-1 to *mir-60*, as suggested by the reviewing editor, and found indeed ISY-1 binds directly to the primary transcript of *mir-60*, based on RNA immune-precipitation of mCherry-tagged ISY-1, followed by QPCR quantification. Furthermore, *mir-60* mutants exhibited elevated expression of *zip-10* (and thereby *asp-17*) and reduced tolerance of CW, thus linking ISY-1 to ZIP-10 regulation both mechanically and functionally. Human ISY1 has been shown to be important for primary microRNA processing (Du et al., 2015). We lend support to this notion and in this work further identify a specific microRNA target that is important for organismic response to CW. We conclude that the ISY-1/*mir-60*/ZIP-10 trio constitutes 3 major players of the novel and likely conserved pathway controlling animal adaptation and response to CW.

If ISY-1 is involved in microRNA processing, then it is probably not specifically involved in CW. Instead it may non-specifically affect this process through processing of microRNAs. If so, loss of other factors that process microRNAs (e.g. dicer, drosha) would be expected to activate the asp-17 reporter. This and whether loss of isy-1 affects microRNA processing should be tested. Interestingly, Figure 4—figure supplement 1 shows that mir-60 is downregulated in isy-1 mutants.

Thank you for raising this important point. We have since confirmed by QPCR that *mir-60* is indeed downregulated in *isy-1* mutants (new Figure 4D). Given our additional new results showing direct binding of *mir-60* to ISY-1 (new Figure 4A and 4B) and the established role of human ISY1 in small RNA processing, we have performed a small-scale candidate RNAi screen and found indeed RNAi against small RNA processing enzymes e.g. *dcr-1* caused *asp-17p*::GFP activation, likely also because of defective *mir-60* processing. We note, however, such RNAi phenotype cannot directly resolve the issue of the specificity of ISY-1 in regulating microRNA; in fact, we found that *isy-1* mutants, unlike *dcr-1* mutants, exhibited down-regulation of only a selective subset of microRNAs including *mir-60* while many microRNAs were actually up-regulated (Figure 4—figure supplement 1). We are collaborating with a small-RNA biochemistry group to characterize how ISY-1 affects small RNA processing specifically for certain microRNA members e.g. *mir-60* and will describe results of such biochemical findings on specificity in a future rather than current genetic study focusing on roles of ISY-1/*mir-60/zip-10*.

ZIP-10 has been previously linked to the innate immunity pathway, and its expression is upregulated upon exposure to bacterial pathogens (Shapira et al., 2006). BATF3, the human ortholog of ZIP-10, has also been linked to immune system regulation and response to pathogens. Is asp-17 upregulated upon exposure to pathogens?

Thank you for raising this interesting question. Expression profiling of a more recent paper (Wong et al., 2007) found indeed up-regulation of *asp-17* by three different pathogens, *E. faecalis, E. carotovora, and P. luminescens*, but their fold induction is relatively moderate (1.84, 1.40, 1.62, respectively) as compared with those by CW or *isy-1* mutations (10~50 fold induction). Compared with *zip-10*, another bZip gene *zip-2* is much more robustly induced by various pathogens with established causal effects (Estes et al. 2010; Shapira et al., 2006). Our results thus support a central involvement of ZIP-10 in the CW response but do not rule out the possibility that ZIP-10 and ASP-17 are also involved (perhaps less prominently) in pathogen response that remains to be resolved in future studies.

Liang et al., 2007 (not cited) showed that zip-10 is expressed in the intestine.What was the rational for focusing on zip-10? Other TFs are also upregulated in isy-1 mutants (jun-1, cebp-1, fos-1, zip-2, and others). Interestingly, zip-2 mediates response to Pseudomonas aeruginosa infection (Estes et al., 2010).The authors propose that ZIP-10 may promote organismal death. Could the authors test if it has such a role in pathogen infection?

We focused on *zip-10* because its deletion or RNAi can completely abolish *asp-17p*::GFP induction upon CW or in *isy-1* mutants (Figure 3) and thus plays a key role in the pathway and biology of interest of current paper. RNAi against other TFs did not have effects in *asp-17p*::GFP induction by CW or in *isy-1* mutants. We agree that ISY-1 likely also regulates other TFs that are involved in processes e.g. pathogen response but unfortunately, we do not have the required biosafety permission yet to perform direct pathogen including *Pseudomonas aeruginosa* experiments in our laboratory.

[Editors' note: the author responses to the re-review follow.]

[…] 1) CW and ISY-1 regulate zip-10 predominantly at the transcriptional level. Because microRNAs can function in the cytoplasm to downregulate targeted mRNAs or inhibit translation, mir-60 may not directly prohibit the transcription of zip-10. It would be of interest to discuss this potential for post-transcriptional regulation in models of of ISY-1and mir-60 may regulate zip-10.

Thank you for the suggestion. We agree and have added discussion texts about the potential mechanisms by which ISY-1 and *mir-60* regulate *zip-10*, indicating that such regulation of *zip-10* is primarily transcriptional based on evidence in this study; further studies are required to discern to what extent *mir-60* might possibly act at the *zip-10* locus or more indirectly impact the transcription of *zip-10, e.g.* by post-transcriptionally inhibiting translation of a transcriptional activator of *zip-10*.

2) In the new Figures 4B-4C, have the numbers been normalized to the amount of immunoprecipitated proteins?

We have used the standard percent input method to measure the RNA-IP signals (https://www.thermofisher.com/us/en/home/life-science/epigenetics-noncoding-rna-research/chromatin-remodeling/chromatin-immunoprecipitation-chip/chip-analysis.html). Samples from each condition exhibit comparable levels of total proteins and ISY-1::mcherry, which was not changed before and after CW based on WB, and were immunoprecipitated by equal amounts of RFP-Trap beads.

3) The legend of panels D & E are missing for Figure 3—figure supplement 1.

Thank you. We have added the legends.

4) In Figure 4—Figure Supplement 1, experimental details and statistics are missing. Including the numbers of fold change in zip-10 mutant would be helpful.

Thank you. We have accordingly added experimental details and statistics.

5) For western-blotting of the proteins, the work used mixed-staged animals. Do ISY-1 and ZIP-10 exhibit stage-specific expression? A population of mixed-staged animals may contain different combination of the stages, which can complicate the explanation.

In Figure 5—figure supplement 2C, we compared the induction of ZIP-10 at different developmental stages and found indeed ZIP-10 is more prominently induced at adult than larval stages, consistent with the enhanced phenoptosis-promoting effect of ZIP-10 in older animals as shown in Figure 5—figure supplement 2D. However, ISY-1 is rather ubiquitously expressed and its RNA or protein level does not appear to be regulated as shown in Figure 2 and Figure 2—figure supplement 2E.

6) In the Materials and methods section, the RNeasy Mini Kit from Qiagen was used to purify total RNAs which are usually longer than 200 nt. How small RNAs were isolated for library construction and deep sequencing needs clarification. Endogenous siRNAs were missed from the Figure 5—figure supplement 1, suggesting that the 5'-dependent method was used to sequence the small RNA library.

We are sorry for the confusion due to inadequacy of our description. In the methods section, the RNeasy Mini Kit from Qiagen was used to prepare total RNA for RNA-seq and QPCR studies. For small RNA sequencing, total RNA was isolated by the Quick-RNA MiniPrep kit (Zymo Research, R1055) that yields total RNA including small RNAs ranging 17-200 nt. We identified endogenous siRNAs in our small RNA-seq datasets but the numbers of reads were small and thus classified as others in Figure 5—figure supplement 1. We have revised the method section to clarify these issues.

7) In Figure 4E, please use different colors to represent different bars. The p-values between zip-10 bars need to be shown. In addition, the relative mRNA level of zip-10 in zip-10(ok3462) is zero and the relative mRNA level of zip-10 in the mir-60; zip-10 double is around 1.5. These results are not consistent. Please explain.

Thank you for the suggestion. We have changed the colors for different bars and indicated p-value significance levels. Given this reviewer’s concern, we have re-performed the experiment and found once again the *zip-10* level is still slightly higher in the *mir-60; zip-10* double than in *zip-10(ok3462)*, indicating that *zip-10* is up-regulated by *mir-60* deletion while *ok3462* likely does not fully delete *zip-10*. This interpretation is consistent with the model in which *mir-60* genetically inhibits *zip-10*.

8) Results section, third paragraph, Figure 2A should be Figure 2B.

Thank you. We have corrected it.